# Sensitivity studies of nighttime TOA radiances from artificial light sources using a 3-D radiative transfer model for nighttime aerosol retrievals

Jianglong Zhang[1], Jeffrey S. Reid[2], Steven D. Miller[3], Miguel Rom_án_[4], Zhuosen Wang[5,6], Robert J. D. Spurr[7], Shawn Jaker[1]

[1]Department of Atmospheric Sciences, University of North Dakota, Grand Forks, ND
[2]Marine Meteorology Division, US Naval Research Laboratory, Monterey, CA
[3]Department of Atmospheric Science, Colorado State University, Fort Collins, CO, USA
[4]Leidos Civil Group, Reston, VA, USA
[5]Earth System Science Interdisciplinary Center, University of Maryland, College Park, MD, US
[6]Terrestrial Information Systems Laboratory, NASA Goddard Space Flight Center, Greenbelt, MD, US
[7]RT SOLUTIONS Inc., Cambridge MA 02138, USA

Corresponding Author: jianglong.zhang@und.edu

**Abstract:** By accounting for surface-based light source emissions and Top-of-Atmosphere (TOA) downward lunar fluxes, we adapted the Spherical Harmonic Discrete Ordinate Method (SHDOM) 3-dimensional (3-D) radiative transfer model (RTM) to simulate nighttime 3-D TOA radiances as observed from the Visible Infrared Imaging Radiometer Suite (VIIRS) Day-Night-Band (DNB) onboard the Suomi-NPP satellite platform. Used previously for daytime 3D applications, these new SHDOM enhancements allow for the study of the impacts of various observing conditions and aerosol properties on simulated VIIRS-DNB TOA radiances. Observations over Dakar, Senegal, selected for its bright city lights and a large range of aerosol optical depth (AOD), were investigated for potential applications and opportunities for using observed radiances containing VIIRS-DNB 'bright pixels' from artificial light sources to conduct aerosol retrievals. We found that using the standard deviation (STD) of such bright-pixels provided a more stable quantity for nighttime AOD retrievals than direct retrievals from TOA radiances. Further, both the mean TOA radiance and STD of TOA radiances over artificial sources are significantly impacted by satellite viewing angles. Light domes, the enhanced radiances adjacent to artificial light sources, are strong functions of aerosol properties and especially aerosol vertical distribution, which may be further utilized for retrieving aerosol layer height in future studies. Through inter-comparison with both day- and night-time Aerosol Robotic Network (AERONET) data, the feasibility of retrieving nighttime AODs using 3-D RTM SHDOM over artificial light sources was demonstrated. Our study shows strong potential for using artificial light sources for nighttime AOD retrievals, while also highlighting larger uncertainties in quantifying surface light source emissions. This study underscores the need for surface light emission source characterizations as a key boundary condition, which is a complex task that requires enhanced input data and further research. We demonstrate how quality-controlled nighttime light data from the NASA's Black Marble product suite could serve as a primary input into estimations of surface light source emissions for nighttime aerosol retrievals.

## 1.0    Introduction

The understanding of diurnal variations of atmospheric aerosols is important for climate, weather, air quality, and visibility analyses and forecasts (e.g. Kaufman et al., 2002; Zhang and Reid, 2010; Hsu et al., 2012; Alfaro-Contreras et al., 2017; Zhang et al., 2021). Although spatial distributions and temporal variations of aerosol particles have been extensively studied using in-situ, ground-based, and passive- and active-based satellite observations during daytime

hours (e.g. Zhang and Reid, 2010; Hsu et al., 2012; Alfaro-Contreras et al., 2017), measurements of atmospheric aerosols during nighttime hours remain limited.

For daytime scenarios, operational aerosol retrievals from reflective solar channels are routinely available from sensors such as Moderate Resolution Imaging Spectroradiometer (MODIS), Multi-angle Imaging SpectroRadiometer (MISR) and Visible Infrared Imaging Radiometer Suite (VIIRS) (e.g. Levy et al., 2013; Hsu et al., 2013; Kahn et al.,

2010). However, there are currently no operational nighttime aerosol data sets available from passive-based satellite observations (e.g. Zhang et al., 2019; Zhang et al., 2008). This is largely because outgoing nighttime visible and shortwave radiation from the moon or surface lights are several orders-of-magnitude smaller than those from reflected and scattered sunlight, making the retrieval of nighttime aerosol retrievals a difficult task (e.g. Zhang et al., 2008; Zhang et al., 2019).

Active space-borne lidar instruments such as Cloud-Aerosol Lidar with Orthogonal Polarization (CALIOP) do provide retrievals of vertical distributions of aerosols for both daytime and nighttime scenarios (Winker et al., 2009). However, CALIOP only provides 2-D curtain views of the earth and atmospheric system with a "beam diameter" of 70 m (Winker et al., 2009) and a total of ~16 narrow-swath orbits per day. Also, non-trivial uncertainties exist in CALIOP-based aerosol extinction and optical depth retrievals as a function of path optical depth; this is due in part to the use

of pre-assumed lidar ratios for given aerosol types (E.g. Midzak et al., 2022) as well as the so-called "retrieval-filled-value" issue, for which aerosol signals as received from lidars are too low for retrievals and thus retrieval filled values are assigned that may introduce sampling-related biases (e.g. Toth et al., 2018).

The VIIRS instrument, present on board the Suomi NPP and NOAA-20 satellites, contains a Day/Night Band (DNB) that can detect upwelling Top-Of-Atmosphere (TOA) radiance from reflected moonlight as well as a host of natural

emission sources (e.g. forest fires, aurora, lightning, and some forms of marine bioluminescence) and artificial light sources on Earth (e.g. cities, gas flares, and ships) (e.g. Schueler et al., 2001; Miller et al., 2013; Wang et al., 2021). In recent studies, both VIIRS DNB observed nighttime light from reflected moon light and from artificial light source emissions were utilized for nighttime aerosol retrievals (e.g. Johnson et al., 2013; McHardy et al., 2015; Zhang et al., 2019; Solbrig et al., 2020). The availability of reflected moon light depends heavily on the lunar cycle, whereas

aerosol retrievals based on artificial light sources can be implemented regardless of the lunar phase (e.g. Zhang et al., 2019). That said, artificial lights have their own host of challenges; most notably, light source characterization, including a target's inherent optical and geometric properties.

To date, aerosol optical depth (AOD) values derived using VIIRS DNB radiances from artificial light sources have been performed by solving 1-D radiative transfer equations in the vertical domain (i.e. 1-Dimensional; Johnson et al.,

2013; McHardy et al., 2015; Zhang et al., 2019). However, artificial light source dependencies (for example, within urbanized areas) are non-trivial in terms of their spatial coverage, temporal intensity, and variation with viewing angle (e.g., Solbrig et al., 2020). Thus, the TOA radiance received from VIIRS DNB includes the attenuated light emission

from a given target as well as diffused/scattered/reflected light from adjacent targets, such as multiple aerosol layers, small clouds, thin cirrus, buildings and ground surfaces, etc.

The complexity of the Earth's nighttime environment is illustrated in Fig. 1, where the top image shows the island of Malta for July 13, 2021 as observed by the Suomi NPP VIIRS DNB (obtained from NASA world view on July 1, 2022; https://worldview.earthdata.nasa.gov/). As depicted, light reaches the sensor from a combination of atmospheric and surface reflected moonlight.  At the same time, surface contributions from various sources (lights or fires) transmit to the top of atmosphere.   In some ways, this environment parallels that for daytime conditions, where the satellite

retrieval must disentangle the surface and atmospheric contributions to the radiance fields, including the contribution of heterogeneous land surface reflectance and adjacency effects therein.   However, there are a number of additional challenges for the nighttime problem.

First and foremost, the light contributions are highly variable (e.g. Cavazzani et al., 2020), by both lunar phase/zenith angle and distance to earth, as well as seasonal and diurnal patterns in light emissions.  For urban environments where

light sources are plentiful, streetlight, traffic and house to urban canyon effects will likely induce a strong view angle dependency for any given source.  Unlike the case with daytime products, where scattering angles are typically in the 110-160$^\circ$ range, for surface lights, scattering angles are much more in the forward scattering direction, where aerosol microphysical sensitivity to the phase function is inherently greater. Thus, while individual light intensity attenuation will follow the Beer-Lambert law, the background light intensity may be phase-function dependent. This is manifested

in nighttime "halo" (or light dome) effects, where surface light emissions are scattered towards the sensor above an unlit surface region. The strength or "contrast" of this dome away from its light source is then also dependent on simultaneous lunar illumination and the height of the aerosol layer.

Taking the above considerations together, nighttime is a truly coupled 3-D radiation problem which is far more complex than its daytime counterpart.  Unlike daytime sensors, this complexity increases further as we explore the

spatial scale dependencies of nighttime radiances – i.e., the observed environment from a 750-meter VIIRS-DNB pixel (observed at heights ~833 km) varies greatly from finer spatial resolution nighttime retrievals (~30 meters, observed at heights < 525km), where the moon-illumination effect on the background scene is significantly reduced. In this study, we have developed a nighttime 3-D radiative transfer modeling (RTM) capability that uses the Spherical Harmonic Discrete Ordinate Method (SHDOM; Evans, 1998) 3-D radiative transfer model, with modification to

include artificial light source emissions, and the replacement of TOA incoming solar radiation with incoming radiation from the moon.  Using these 3-D nighttime RTM simulations, we have studied the sensitivity of VIIRS DNB radiances to atmospheric and lower boundary conditions, and investigated the feasibility with the use of VIIRS DNB data for retrieving aerosol properties such as AOD and aerosol layer height.  In addition, we have assessed the utility of monthly NASA Black Marble products generated from daily lunar bidirectional reflectance distribution functions

(BRDFs) corrected radiances from nighttime light sources as a possible lower boundary condition for aerosol retrievals (Román et al., 2018; Wang et al., 2022).

This paper is organized as follows: In Sect. 2 datasets and models used in the study are discussed; in Sect. 3 sensitivity studies of nighttime aerosol retrievals for a range of observing conditions are presented; in Sect. 4  the feasibility for

using the Black Marble data as surface light source emissions for nighttime 3-D RTM calculations is discussed. Sect. 5 summarizes the work and offers some concluding remarks toward future research.

### 2.0      Datasets and Models

### 2.1 VIIRS data

With inheritance from the Operational Linescan System (OLS) sensor on board the Defense Meteorological Satellite Program (DMSP), the VIIRS DNB observes nighttime TOA radiances in the visible to near-infrared spectrum (0.5 - 0.9 μm) at a peak wavelength of ~0.7 μm and a spatial resolution of ~750 m (Schueler et al., 2001). The VIIRS DNB can detect nocturnal low light from earth scenes that is "10 million times fainter than reflected sunlight" (Miller et al., 2013), which enables observation of nighttime lights that are not detectable from traditional passive-based sensors on low earth orbit such as MODIS, AVHRR and MISR. Compared to OLS, which is not calibrated and provides a form of 'constant contrast' imagery via a gain factor applied in a complex and non-linear way, the VIIRS DNB radiances are well calibrated, with reported calibration uncertainties on the order of 2-6% for Suomi NPP VIIRS DNB (Chen et al., 2017). In this study, Suomi NPP VIIRS data from 2019 were used. The VIIRS calibrated low light radiances were obtained from the VIIRS Sensor Data Record (SDR) for DNB band (SVDNB) data. Geolocations for VIIRS observations were obtained from the VIIRS DNB SDR Ellipsoid Geolocation (GDNBO) data. We used the VIIRS Summed Cloud Cover (VCCLO) data for cloud clearing of observed scenes.

We selected Dakar, Senegal (14.72 °N, 17.47 °W) as our study region, as it is a coastal city in North Africa that experiences dominantly African dust nearly year-round. To enable comparisons with 3-D RTM model output, VIIRS observations around Dakar were averaged into 0.742x0.742 km$^2$ bins with a total of 35 x 35 bins covering the study area. Fig. 2a shows the study region for a moonless night on November 28, 2019. Regions far removed from the city light sources are visibly dark. Spatial inhomogeneity is observed; bins near commercial centers contain radiances exceeding $1 \times 10^{-7}$ W cm$^{-2}$ sr$^{-1}$, while the averaged radiance over the artificial light sources is around $3 \times 10^{-8}$ W cm$^{-2}$ sr$^{-1}$.

Cloud contamination presents a significant issue with nighttime observations, especially for optically thin clouds which can scatter/diffuse surface based light emissions. Thus, besides using cloud flags from the VCCLO data to exclude cloudy nights, each VIIRS granule over Dakar for 2019 was also visually inspected for any residual cloud contamination. This process resulted in 91 cloud-free granules for 2019 over Dakar that were used in the study. The list of the 91 granules is included as supplemental data.

### 2.2 NASA's Black Marble Product Suite

The VIIRS/NPP Lunar BRDF-Adjusted Nighttime Lights Monthly Level3 data (VNP46A3/VJ146A3) contains mean monthly nighttime lights (NTL) radiance values with outliers removed for multiple view angle and snow status categories along with ancillary datasets (quality flags, standard deviation, number of clear observations, land water mask, and platform; Wang et al., 2022). The monthly NTL data are made available in 2400x2400 pixel grids (or "tiles"), making up a global-coverage grid of ~431 10°x10° tiles (excluding oceans) at a resolution of 15 arc-seconds.

The monthly NTL data from the VNP46A3 product are generated from atmospheric and lunar BRDF effect corrected radiances (Román et al., 2018; Wang et al., 2022) using the lunar irradiance model of Miller and Turner (2009). We use this dataset to describe surface nighttime emissions for our current study. The nighttime surface emission sources for the 3-D RTM runs are in 35x35 bins with a bin size of 742 meters. Thus, the nearest-neighbor method was used for constructing Black Marble product-based surface light emission sources, based on the surface emission sources developed from this study (Fig. 2b) on a monthly basis. These monthly derived Black Marble surface light emission sources were further used for estimating surface light emission sources over Dakar for this study.

## 2.3 AERONET data

Aerosol Robotic Network (AERONET) AOD data were examined for the selection of relatively aerosol free nights and for evaluation of 3-D RTM-based aerosol retrievals. Through measuring attenuated solar light at ground stations, AOD data were derived at several spectral channels ranging from 340nm – 1640 nm, with an uncertainty of the order of ~0.01 (Holben et al., 1998). To evaluate nighttime VIIRS AOD retrievals, Version 3, Level 2 quality-assured daytime AERONET data at 675 nm over Dakar (14.394°N, 16.959°W) were chosen. To estimate nighttime AERONET data from daytime AERONET data for a given day, daily averaged AERONET data from the previous day and the current day were obtained. If the AOD values from the two daily averages are less than 0.2, indicating a lower variation in AODs for the two days (and possibly, the intervening nighttime hours), then AOD values from the two days were averaged to represent the daytime AOD value for the intervening night.

Through observation of attenuated direct-beam moonlight, nighttime AERONET AOD retrievals were also available from AERONET, but of course only for nights with available moonlight. Version 3, Level 1.5 nighttime AERONET AOD at 675 nm over Dakar were also used to spatially and temporally collocate with VIIRS AOD retrievals from this study. Only nighttime AERONET data within ± 30 minutes of VIIRS overpass times were incorporated in the averaging procedure.

## 2.4 SHDOM model

The SHDOM model (Evans, 1998) is a 3-D RTM that simulates and outputs 3-D radiation fields in terms of both radiances and fluxes for user-specified observing conditions. The SHDOM solves for the 3D radiation field using discrete ordinate methods based on the representation of parts of the source term in terms of spherical harmonic expansions. Publicly available, SHDOM has been extended to include linearization (Doicu and Efremenko, 2019); it has also been used to study cloud-adjacency effects on aerosol property retrievals (Wen et al., 2016),
SHDOM is designed for daytime applications that require TOA solar radiation as an input. For simulating nighttime artificial light sources, we have modified SHDOM to include nighttime-related boundary (surface and TOA) conditions. Firstly, nighttime surface artificial light emissions are included as inputs for surface conditions; we have assumed that artificial light sources generate Lambertian emissions with no azimuth dependence. In reality, artificial light sources are a strong function of viewing angle (e.g. Johnson et al., 2013; McHardy et al., 2015; Zhang et al., 2019) due in part to the urban canyon effect, light shielding, and other geometric factors. Thus, we have implemented a correction for the sensor viewing angle, described in the section to follow. In reality, surface light emissions typically

are heterogeneous. However, quantifying the heterogeneity of surface emissions is a significant research undertaking requiring additional study beyond the scope of this investigation.

SHDOM is a grid-based model and surface properties at non-grid locations are interpolated through bilinear interpolation. This interpolation scheme can create a problem, as nighttime emissions from artificial light sources are spatially inhomogeneous (e.g. an example of surface emission for Dakar is shown in Fig. 2a; the method for constructing the surface emission sources is discussed further along) and bilinear interpolation may result in unrealistic surface emission representation. Thus, the nearest-neighbor method was applied here to ensure surface emission

values were not overly perturbed.

Secondly, we have replaced downwelling TOA solar inputs with lunar irradiance parameters: the lunar zenith angle replaces solar zenith angle, and TOA downward flux from the moon replaces the TOA solar flux. Lunar phase dependent TOA downward fluxes are provided by the Miller-Turner lunar irradiance model (Miller and Turner, 2009). Surface light source emissions were derived from VIIRS DNB data; spectrally dependent surface light source

emissions are not available. Thus, SHDOM simulations were performed at the 700 nm spectral channel, which is the peak wavelength for VIIRS DNB. The VIIRS DNB filter response function was used to weight TOA downward lunar fluxes (centered at 700 nm), similar to the procedure mentioned in Miller and Turner (2009). Rayleigh scattering from air molecules is included (Evans, 1998; Fu and Liou 1992), while trace gas molecular absorption (e.g. $H_2O$) is taken to be negligible in the 700 nm channel (similar to the VIIRS DNB; e.g. Wang et al., 2016). Hereafter, AOD refers to

the aerosol optical depth at 700 nm unless specifically mentioned.

We adopted the SHDOM Property File Generation (PROGEN) System for generating aerosol optical properties in this study. We assumed that the Dakar region is dominated by dust aerosols year round. For simplicity, dust particles were assumed as spheres with a log-normal particle size distribution. The mean radius was set to 1.75 μm (or mean diameter of 3.5 μm) and the geometric standard deviation was set at 1.0 μm (Reid et al., 2008). The dust bulk aerosol

density was taken to be 2.5 g/cm$^3$ (Reid et al., 2008). Note that the spherical-particle assumption is necessary here, as the PROGEN system does not yet include handling of dust non-sphericity. Also, dust aerosol properties vary as a function of dust morphology (e.g. Conny and Ortiz-Montalvo, 2017). The handling of dust non-sphericity is left for future investigation. The refractive index of dust aerosol at 700 nm is taken to be 1.51-0.002 for the Sahel region (Di Biaggio et al., 2019). The vertical distribution of dust aerosols (shown in a later figure, Fig. 8a) is peaks at 1 km as

suggested from Mortier et al., (2013), in which aerosol properties from Dakar were studied using combined micro-lidar and AERONET data.

To simulate VIIRS data given in 35×35 sample bins, we have added 10 bins in each direction; these additional bins are assumed to be dark surfaces. The reasoning behind this addition is the requirement to study the enhancement in radiance adjacent to artificial light sources, as mentioned in Sect. 3.4. Thus, the study domain is actually 55×55 bins

in size, with the outer 10 bins on each side assumed to be free of artificial light sources.

## 2.5 Construction of surface emissions for 3-D RTM simulations

Surface emissions from artificial light sources are needed as inputs for the SHDOM mode simulations. However, as noted from previous studies (e.g. Zhang et al., 2019; Wang et al., 2021; Solbrig et al., 2020; McHardy et al., 2015),

surface light source emissions are a strong function of viewing zenith angle (VZA). Therefore, a total of 34 cloud-free nights (the list of those nights is included in the Supplement to the paper), with relatively low daytime/nighttime AERONET AODs of around 0.2 or less, were chosen to study the relationship between VZA and TOA radiance, as shown in Fig. 3. For each night, for all 1225 VIIRS bins, we excluded 0.5% bins with the highest radiance values (brightest) and 75% bins with the lowest radiance values (darkest). The 75% darkest bins are likely background bins. We found that large temporal variations sometimes exist in the brightest bins, and we have also excluded these bins from the study. The remaining bins were averaged to give the mean TOA radiances from Dakar for a given night. Normalized TOA radiances are computed by dividing the mean TOA radiances from each night with the averaged mean TOA radiance values from all 34 nights.

As shown in Fig. 3, a strong correlation exists between the normalized TOA radiance and VZA over Dakar for the above mentioned 34 nights. For cosine(VZA) values close to 1.0 (i.e., viewing nearly straight down on a target), a large spread in normalized radiances was found, ranging from 0.84 to 1.13. For cosine(VZA) ranging from 0.45 to 0.95 (i.e., viewing the target from more oblique angles), normalized radiances ($I_{normalized}$) were found to decrease monotonically, that can be approximated with linear regression:

$$I_{normalized} = 1.34 - 0.40 \cos (VZA).$$                    Eq. 1

To derive surface emissions, and to account for the VZA effect, only 19 cloud-free nights with relatively low aerosol loading (either day or nighttime AODs of ~0.2 or less) and with cosine (VZA) larger than 0.8 (biased to near-nadir viewing) were selected. A list of those selected nights is also included in the Supplement to the paper. Eq. 1 is applied as a first-order representation of the relationship between VZA and radiance. In addition, the average daytime/nighttime AERONET AOD for the 19 cloud-free and relatively aerosol-free nights is around 0.2. Surface emissions represent nighttime emissions from artificial light sources seen on relatively aerosol- and cloud-free nights. Thus, 3-D RTM simulations were performed twice, for moonless nights (no TOA moon flux) at nadir view (VZA zero) for AOD values of 0 and 0.2 respectively. Ratios of 3-D RTM simulations for bins with artificial light sources were used to compute surface emissions from cloud- and aerosol-free nights from the averaged radiance values over the chosen 19 nights.

The derived surface emissions map is shown in Fig. 2b and the standard deviation (STD) of surface emissions from the 19 nights is shown in Fig. 2c. The derived surface emissions show a pattern similar to TOA radiances as seen from Nov. 28, 2019 (Fig. 2a). This is not a surprise, since the nighttime aerosol loading was expected to be relatively low on Nov. 28, 2019; indeed, an averaged daytime AERONET AOD of around 0.14-0.17 was observed for the two adjacent days. Still, large variations are found for regions with bright light sources, as shown in Fig. 2c. The peak STD of radiances (from temporal variations) for the artificial light sources is $1.28 \times 10^{-7}$ W cm$^{-2}$ sr$^{-1}$, with the mean STD of $7.05 \times 10^{-9}$ W cm$^{-2}$ sr$^{-1}$. The peak STD values from the temporal variants are found over peak surface emission regions as shown in Fig. 2b. This STD behavior indicates that bins with the brightest surface emissions contain large temporal variations and are not suitable for use in aerosol retrievals. Thus, in the following investigations, we have excluded 0.5% of the brightest bins and 75% darkest bins (for the inner study domain of 35 x 35 bins), with the remaining bins used for aerosol retrievals and for sensitivity studies.

**2.6 Inter-comparison of 1-D with 3-D RTM runs**

To evaluate/validate the developed 3-D nighttime RTM capability, we inter-compared both day- and night-time simulations from this study with Spherical Harmonic Discrete Ordinate Method for Plane-Parallel Atmospheric Radiative Transfer, or SHDOMPP (Evans, 2007) model simulations. SHDOMPP is an unpolarized 1-D RTM for simulating radiation fields at daytime. Similar to SHDOM, source terms are represented in terms of spherical harmonic expansions; details of SHDOMPP can be found in Evans, (2007). SHDOMPP was validated against the Discrete Ordinate Radiative Transfer Model (DISORT) in a past study for daytime applications (Evans, 2007), and was chosen as it takes similar inputting parameters (e.g. atmospheric and surface properties) as SHDOM. Thus data preparation steps are rather simple.

We also enhanced SHDOMPP with a capability for simulating nighttime artificial light sources using similar approaches as described in Sect. 2.4 and 2.5. To be specific, the Miller and Turner (2009) lunar irradiance model was incorporated to provide estimations of TOA lunar flux and a surface artificial light source emission was included as a lower boundary condition.

First, we inter-compared simulated single column TOA radiance for daytime conditions from both the 1-D and 3-D RTMs as shown in Fig. 4a at two aerosol loading scenarios (AOD =0, and AOD =1.0 for AOD at 700 nm) using the dust aerosol model as described in Sect. 2.3. The simulations were carried out for cosine VZA rages of 0.4 to 1.0, and with a normalized TOA downward solar flux of 1. For the single column simulations, the 3-D radiative transfer model essentially runs with a study domain of 1x1 grid (or the study domain of the 1-D runs). As shown in Figure 4a, for daytime simulations, TOA reflectance from the 1-D and 3-D radiative transfer models match closely, with a correlation and a slope of near 1 for both Rayleigh sky (AOD =0) and dust polluted sky (AOD =1.0). This suggests that daytime simulations from both 1-D and 3-D RTMs are consistent.

Also, for a given zenith angle (solar zenith angle for the TOA downward path and sensor zenith angle for the surface upward path), and for a given atmospheric condition, the surface downward radiance/flux (only TOA downward solar flux with no surface artificial light source emission) shall go through the similar radiative processes as the TOA upward radiance/flux (only surface artificial light source emission with no TOA incoming solar/moon flux). Thus, by using the same set of cosine zenith angles (ranging from 0.4 to 1.0), for two aerosol loading scenarios as used in Fig. 4a, we inter-compared daytime surface downward radiances (include both direct and diffuse radiation, assuming normalized TOA inputting solar flux of 1, no surface artificial light sources, and a dark surface) and nighttime TOA upward radiances (assuming normalized surface light source emission flux of 1, no TOA solar or moon flux and a dark surface) from the single column 3-D RTM runs as shown in Fig. 4c. A near perfect 1 to 1 relationship is found between the two variables, suggesting the nighttime processes function as designed.

We also inter-compared the single column simulations at nighttime form both 1-D and 3-D RTMS as shown in Fig. 4b. Here, we used a normalized Lambertian nighttime artificial emission flux of 1 for both 1-D and 3-D runs and for two aerosol loading scenarios and for viewing angle ranges as used in constructing Fig. 4a. Again, near perfect matches in normalized TOA radiances are found between 1-D and 3-D simulations (Fig. 4b).

Upon validating of the newly developed 3-D nighttime modeling capability using the 1-D RTM for single column simulations, both the 1-D and 3-D radiative transfer models were also applied to simulate VIIRS nighttime radiances over Dakar for multi-column simulations for a study domain with 3025 (55x55) grid points. To perform the multi-

column simulations using the 1-D radiative transfer model, 3025 single column simulations were performed, without considering contributions from adjacent grid points as is the case for the 3-D simulations. As shown in Fig. 4d (Nov. 28, 2019 over Dakar), the 1-D radiative transfer model runs match reasonably well with 3-D runs for relatively dim light sources, yet simulated radiances are much higher over very bright light sources for the 1-D runs. Also, for regions with no artificial light sources, the simulated radiance is 0 for 1-D runs, and is non-zero for the 3-D runs due to the adjacency effect.

Also, simulated standard deviations of radiances from both 1-D and 3-D nighttime radiative transfer models were inter-compared with observations from VIIRS for 32 cloud-free nights (that have collocated nighttime AERONET measurements) from 2019 over Dakar (Fig. 4e). Here, aerosol loadings were obtained from the collocated nighttime AERONET AOD data. The RMSE error in simulated standard deviations from the 1-D runs is nearly twice as that for runs from the 3-D simulations; this suggests that the adjacency effects need to be considered for simulating TOA nighttime emissions over artificial light sources.

### 3.0    Sensitivity study & discussion

Simulated TOA radiances from VIIRS were examined for different observing conditions and aerosol properties. We evaluated changes in simulated TOA radiances for both moonless and moonlit nights as functions of VZA and aerosol loading. Nonetheless, TOA downward lunar fluxes change markedly as a function of lunar phase, and have a strong night-to-night variation. Thus, for other sensitivity studies including the study of impacts of aerosol vertical profile, aerosol properties and azimuth angles, only moonless night simulations were considered, so that impacts of selected variables controlling simulated TOA radiances can be isolated from the influence of moonlight.

#### 3.1 Sensitivity as a function of viewing angle

The sensitivity of TOA radiance as a function of VZA is studied for both moonless and moonlit nights. For a moonless night, the TOA moon flux is set to zero. For a moonlit night, the (VIIRS/DNB spectral response function-weighted, integrated over the band) TOA downward lunar flux is set to $2.37 \times 10^{-8}$ W cm$^{-2}$ with a cosine lunar zenith angle of 0.412. The choices of TOA downward lunar flux and cosine lunar zenith angle is to some extent arbitrary (based on similar values from a randomly selected night of Dec. 18, 2018), since both variables vary on a nightly basis depending on lunar phase and zenith angle.

For both moonless and moonlit nights, SHDOM model simulations were run for cosine VZA varying from 0.4 to 1.0 with an interval of 0.1, and for AOD varying from 0 to 1 with an interval of 0.1. These simulations are shown in Fig. 5 for both the mean radiance and STD of radiances of artificial light sources. Here, the STD represents the spatial variance of radiances over artificial light sources, and we carry the same meaning hereafter unless specifically mentioned. Both the mean radiance and STD of radiances decrease with increasing AOD. This is not a surprise, since, in the presence of an aerosol layer, surface emissions are attenuated and the contrasts among artificial light sources are reduced (e.g. Zhang et al., 2019). Also, sharper decreases can be seen for lower cosine VZA values (longer slant paths), indicating that VIIRS observations are more sensitive to aerosol loading for higher viewing zenith angles.

For moonless and moonlit nights, there are systematic differences in simulated radiances as a function of AOD over artificial light sources. As one would expect, for moonlit nights mean TOA radiances increase with AOD, similar to the case for daytime conditions and solar path radiances. By comparison, no changes are found between the relationship of STD of radiances and AOD for both moonless and moonlit nights. This is not a surprise as TOA downward moon flux is assumed to be the same for all grids within the study domain, and thus adding moon lights would not affect STD of radiances. Given that lunar models still contain non-negligible (5-15%, depending on lunar phase) uncertainties, this exercise suggests that the STD of TOA radiances is a better parameter for intercomparison with the mean TOA radiance for nighttime AOD retrievals.

**3.2 Sensitivity as functions of sensor azimuth angle.**

Here we have examined the impact of sensor azimuth angle on TOA radiances, for moonless nights. Also, since the lunar azimuth angle is only important for moonlit nights and surface emissions are independent of moon status, and thus satellite azimuth is set to 0°. As suggested from Figs. 6a and 6b, for moonless nights, both the mean and STD of radiances as a function of AOD are nearly independent of sensor azimuth angle. Even for observation scenarios with cosine VZA of 0.5 (Fig. 6b), both the mean TOA radiance and STD of TOA radiances are still only weakly dependent upon sensor azimuth angle. Again, this phenomenon is largely due to our Lambertian azimuth assumption for surface emissions. Still, for the limited scope of this study, we assume sensor azimuth angle to 0° in our simulations to follow.

**3.3 Sensitivity as functions of refractive index of dust aerosols**

One of the key aerosol optical properties is the refractive index (RI), which defines the scattering and absorbing characteristics of aerosol particles. Although the real and imaginary parts of the dust aerosol refractive index were set to (1.51, -0.002i) for the Sahel region according to (Di Biaggio et al., 2019), the STDs of these components were set to 0.03 and 0.0014, respectively, based on collected soil samples for the Sahel region. Thus, the impact of refractive index of dust aerosols on simulated TOA radiances were studied by varying the real part of the RI from 1.48 to 1.54 and the imaginary part from 0i to -0.004i.

Figures 6c and 6d show the changes in the mean TOA radiance and STD of radiances as a function of RI, for moonless nights and for cosine VZA values of 1.0 and 0.5. Although variations in the mean TOA radiances due to the changes in RI increase as AOD increases, the variations are marginal. It is also important to note that for moonless nights, the STD of radiances for artificial light sources are less dependent of changes in RI; once again, this suggests that STD of radiances is a better variable than the mean TOA radiance for AOD retrievals.

**3.4 Sensitivity as functions of light domes**

In addition to direct observation of artificial lights from VIIRS data, light domes, which are caused by diffusely scattered light in the vicinity of artificial light sources, are also observable in the VIIRS measurements. These light domes contain information about optical properties of aerosol particles, and they could be used for retrieving aerosol optical depth and/or physical properties. However, we do not attempt such a retrieval here.

To investigate the behavior of light domes as functions of observing conditions, and to exclude the effect of moonlight, we have run simulations in moonless conditions for different aerosol loadings and for different VZA values. Fig. 7a shows the simulated light dome pattern for the AOD = 0 case, where the red bins shown in the plot are masked surface emission sources. No apparent domes are observable for the AOD = 0 case, which is not surprising as Rayleigh scattering is insignificant at 700 nm. In contrast, domes are observable for AOD = 1.0 for cosine VZA of 1.0, as shown in Fig. 7b. We further quantified domes as a function of the spatial distance from any artificial light sources, as shown in Figure 8a. To construct this figure, radiance values for bins that are 1-15 bins distance from any adjacent artificial light sources are averaged. For the cosine VZA = 1.0 case, 1 bin away from any known light sources, the averaged radiance is ~$1.16 \times 10^{-10}$ W cm$^{-2}$ sr$^{-1}$ for the aerosol-free case and around $1.72 \times 10^{-9}$ W cm$^{-2}$ sr$^{-1}$ for AOD = 0.5. The near 15-times increase in radiance value is due to diffusely scattered light from the aerosol layer.

For the cosine VZA =0.5 case (Fig. 8b), for a bin that is just one bin distant from any known light sources, the averaged radiance is ~$1.91 \times 10^{-10}$ W cm$^{-2}$ sr$^{-1}$ for aerosol-free cases and is around $3.67 \times 10^{-9}$ W cm$^{-2}$ sr$^{-1}$ for AOD = 0.5. The enhanced radiance in the vicinity of artificial light sources may be due to larger viewing angle and longer slant path. However, it is also possibly influenced by uncertainties due to the viewing angle regression, as noted in Eq. (1) above. In general the light dome effect diminishes rapidly for 7-8 bins of more away from artificial light sources. As illustrated in Figs. 7 and 8, enhancements in radiance in the vicinity of artificial light sources contain aerosol information that can be used for future aerosol property retrievals, possibly using 3-D RTM. This is because it is difficult for 2-D RTMs to accurately account for scattered lights originated outside the targeted 2-D domain. .

### 3.5 Sensitivity as a function of aerosol vertical distributions

We also examined the impact of aerosol vertical distribution on TOA radiances and light domes. This exercise was performed by repositioning the peak aerosol plume height from 1 km (e.g., Mortier et al. 2013) to 2 km. Figure 9a shows as an example the vertical distributions of dust aerosol concentration for AOD of 0.1 before and after the modification of the aerosol vertical distribution. As in previous tests, scenarios are moonless nights with no downward lunar flux. As shown in Fig. 9b, with increasing elevation of the dust plume both the mean radiance and STD of radiances were reduced. For cosine VZA of 1.0, a 7% and 12% reduction in the mean radiance and STD radiances, respectively, were found for AOD of 1.0. For cosine VZA of 0.5, the simulated mean radiance reduced from $1.3 \times 10^{-8}$ W cm$^{-2}$ sr$^{-1}$ to $1.1 \times 10^{-8}$ W cm$^{-2}$ sr$^{-1}$ and the simulated STD of radiances reduced from $5.5 \times 10^{-9}$ W cm$^{-2}$ sr$^{-1}$ to $3.9 \times 10^{-9}$ W cm$^{-2}$ sr$^{-1}$ by moving the dust plume with an AOD of 1.0 from a peak altitude of 1 km to a peak at 2 km.

In contrast to the earlier result indicating a reduction in light dome radiance, an increase in light dome size was observed when the peak dust plume was elevated from 1 km to 2 km. As suggested from Fig. 8c, for the elevated dust plume with cosine VZA of 1.0, enhancements in radiances can be seen as much as 10-11 bins away from surface light sources, with a ~30% increase in light dome size compared to the situation in Fig. 8a, where the peak aerosol plume height was set to 1 km. Similar effects were also found with cosine VZA of 0.5, where the light dome size increased from 8 bins to 12-13 bins separation from artificial light sources when the plume height was elevated from 1 to 2 km (Figs. 8b and 8d). These simulations show how light dome size, due to the effects of aerosol scattering and diffusion,

is a strong function of aerosol vertical distribution. This insight may be useful for retrieving nighttime aerosol vertical distributions in the future, especially as observations from light domes typically are not used for AOD retrievals.

**3.6 Experiments with different aerosol vertical distributions**

As noted in the previous section, VIIRS simulated TOA nighttime radiances over/near artificial light sources are a strong function of aerosol vertical distribution. These simulations assumed a vertically homogenous aerosol layer. Extending the analysis of the previous section, we also conducted two experiments with different scenarios.

For the first experiment, we assumed the presence of two vertically inhomogeneous aerosol layers. The first layer is the background aerosol layer, with an AOD of 0.1. Aerosol concentrations decrease exponentially with an arbitrary scale height of 1.2 km (Black color curve as shown in Fig. 10a). The assumed AOD for the second layer is 0.4 and we moved this layer to 1, 3, 5 and 7 km to examine the sensitivity of the aerosol layer height to VIIRS TOA radiance over/near artificial light sources (e.g. Fig. 10a).

For the second experiment, we assumed the presence of a single aerosol layer with a Gaussian distribution as shown in Fig. 10b. Again, the total AOD for this Gaussian layer is 0.5, which is the same as that for the first experiment, and we set the peak height to 1, 3, 5 and 7 km (with standard deviation of 0.5, 1.5, 2.5 and 3.5 km) for the sensitivity analysis. Also, to study the enhanced radiances for areas away from the artificial light source regions and because aerosol plumes are rather elevated for these two experiments, we added 10 more bins to each side of our 55 x 55 domain, making for a field of 75 x 75 bins in size.

For the first experiment, by moving the second aerosol layer from 1 to 7 km, decreases in both TOA radiances and STDs of radiances over the artificial light sources were found, as shown in Fig. 10c for cosine VZA ranging from 1.0 to 0.4. Correspondingly, enhancements in both light dome size and radiance intensity over the light dome region were observable with the increase of aerosol plume height for the second aerosol layer (e.g., Fig. 10e for the nadir view case). In addition, larger variations in light dome patterns are found for the second aerosol layer located at lower altitudes such as 1 and 3 km, suggesting light domes are more sensitive to aerosol layer changes at lower altitudes.

Although aerosol distributions from the second experiment (Fig. 10b) are different from those of the first experiment (Fig. 10a), it is perhaps surprising to see that the averaged radiances, STDs of radiances over the artificial light sources regions (Figs. 10c, 10d), as well as the light dome patterns (Figs. 10e and 10f), are similar for the same peak aerosol plume altitude. This finding suggests that TOA radiances as observed from VIIRS DNB near or over the artificial light sources are a stronger function of the peak aerosol layer height than of the specific shape of aerosol vertical distribution.

**4.0 Feasibility use of 3-D RTM for aerosol retrievals**

In this section, we evaluated the feasibility of using 3-D RTM simulations for AOD retrievals through the use of both day- and night-time AERONET data. In addition, a prior knowledge of artificial light source emissions are needed for nighttime AOD retrievals over artificial light sources, and thus, we also studied the potential of using atmospheric and lunar corrected monthly averaged nighttime radiances from NASA's Black Marble products as surface light source emissions in this study.

**4.1 3D-RTM-based nighttime aerosol retrievals vs. ground-based AERONET data**

Using 91 cloud-free nighttime observations from VIIRS DNB over Dakar, we have examined the feasibility of retrieving AOD using 3-D RTM simulations. The surface emissions as shown in Section 2 are used as the artificial light sources, after accounting for the VZA dependency as shown in Eq. 1 and Fig. 3. The VIIRS viewing geometries and moon phases were used for computing incoming moon fluxes. For each night, SHDOM model runs were performed for AOD ranging from 0 to 1 with an interval of 0.1 (a total of 11 entries) for given observing conditions such as viewing geometry and lunar phase angle. Surface albedo was assumed to be 0.1 (Jäkel et al., 2013).

As suggested by the results of our sensitivity study (Sect. 3), the impact of azimuth angles is sufficiently small to fix the sensor azimuth angle at 0° without incurring large errors. Additionally, since our sensitivity studies indicated that the STD of the radiances is less sensitive to TOA downward moon flux, we used the STD values for AOD retrievals. In fact, the use of STD of radiances as a tool for AOD retrieval has been suggested already from several previous studies (e.g. Zhang et al., 2019; McHardy et al., 2015). To perform AOD retrievals, simulated STD of radiances from 11 different AOD values were inter-compared with STD of radiances from VIIRS data, and the AOD value was retrieved by finding the best matching AOD that generates the same STD of radiances as the observation through linear interpolation.

Figures 11a and 11b show comparisons of AERONET versus VIIRS AOD, using nighttime and daytime AERONET data, respectively. A correlation of 0.5-0.6 was found between AERONET and VIIRS AOD when using both day and nighttime AOD, with root mean square errors of 0.16-0.20, suggesting that the proposed method held qualitative skill. One of the major sources of uncertainty is the surface emission. As seen in Figs. 2b and 2c, the STD of surface emissions (temporal changes) is around 25% of the mean surface emission on average, suggesting there are temporal or view angle variations in artificial light source patterns. A weekly- or monthly-based surface emission database may be needed for accurately retrieving nighttime AOD using artificial light sources through 3-D radiative transfer model simulations.

As suggested from our sensitivity study in Sect. 3, the STD of radiances is more sensitive to AOD at larger viewing angles. Accordingly, we repeated the exercise from Figs. 11a and 11b, but now restricting cosine VZA to values less than 0.8; the results are shown in Figs. 11c and 11d. Improved correlations of 0.62 and 0.71 were found for comparisons between VIIRS AOD and nighttime and daytime AERONET AOD data, respectively, associated with reductions in RMSE values. This improvement suggests that observations from higher viewing angles can be better used for nighttime AOD retrievals using artificial light sources.

**4.2 Feasibility of using the Black Marble products for night time aerosol retrievals**

We also investigated the feasibility of using NTL data from the NASA Black Marble products (Román et al., 2018; Wang et al., 2022) as surface light source emissions for 3-D RTM-based nighttime aerosol retrievals. Figure 12a shows the yearly mean NTL data, constructed using averaged monthly Black Marble NTL data, and subsequently sorting monthly Black Marble NTL data into surface light mission sources as developed from this study (Fig. 2b) using a nearest-neighbor method. The yearly mean NTL patterns as shown in Fig. 12a are in close similarity with the background light emission sources constructed in our study (Fig. 2b); this is reinforced with the high correlation of

0.95 between the two datasets (Fig. 12c). However, larger discrepancies were seen over very bright spots, with the Black Marble NTL values above $7 \times 10^{-8}$ W cm$^{-2}$ sr$^{-1}$, indicating larger variations exist over regions with the brightest light source emissions (e.g. Fig. 12b); these regions need to be excluded for nighttime aerosol retrievals using artificial light sources.

Interestingly, a correlation of near 1 (0.99) was found between the retrieved nighttime AODs using the surface light source emissions estimated from this study and the Black Marble NTL data. This indicates that the year-mean Black Marble NTL data could be directly used as the surface light emission sources for future studies. Although we also tried to use monthly Black Marble NTL data as monthly-based light emissions sources for aerosol retrievals, we found that larger discrepancies existed between VIIRS and AERONET AODs using this method (results not shown). This could be due to the strong VZA dependency in light emission sources; in the future, monthly Black Marble NTL data may need to be revised to better account for VZA dependency.

**5.0     Conclusions**

For this research we modified SHDOM, a 3-D radiative transfer model (RTM), by adding nighttime surface emissions from artificial light sources and replacing TOA solar radiation with downward fluxes from the moon. In this way we have developed a 3-D RTM capability for simulating TOA radiances from artificial light sources for aerosol retrievals. Using VIIRS data over Dakar, Senegal, we have performed sensitivity studies and we have examined the feasibility for the use of 3-D RTM simulations for nighttime aerosol retrievals. Our conclusions are listed as follows:

- Large night-to-night variations were found in emission from artificial light sources over Dakar. While cloud-free sky surface emissions were constructed using VIIRS data from 19 cloud-free and relatively aerosol-free nights, the standard temporal deviation (STD) of nighttime artificial light emissions was around 25% of the mean averaged TOA radiance from artificial light sources. These temporal changes may introduce non-trivial uncertainties in aerosol retrieval using artificial light sources, and hence estimations of artificial light source emissions may need to be constructed for shorter temporal windows in future studies.

- Consistent with previous studies (e.g., Zhang et al., 2019; Solbrig et al., 2020), TOA radiances from artificial light sources vary strongly with sensor viewing angle, indicating that such sources are often heterogeneous. We found that both the mean TOA radiance and STD (spatial variation) of TOA radiances from artificial light sources are more sensitive to aerosol loading at larger viewing angles. Lesser impacts were found from variations of aerosol optical properties (refractive index) and azimuth angle.

- Also, larger variations in simulated mean TOA radiances from artificial lights were found when compared with variations in the STDs of TOA radiances as a function of aerosol optical depth (AOD), suggesting that the STD of TOA radiances is a more reliable parameter for AOD retrieval using artificial light sources.

- Enhancements in radiances in the vicinity of artificial light sources (the so-called light dome effect) are strong functions of aerosol properties. Light dome size increases as aerosol plume height increases, indicating that these domes may be used for deriving aerosol vertical distributions using VIIRS DNB data. We also found that light dome patterns are a stronger function of peak aerosol plume height than the shape of aerosol vertical distribution; further, changes in light dome patterns are more sensitive to aerosol vertical distribution at lower altitudes. In addition, although our discussions have focused on aerosols, similar methods may be applicable for cloud property retrievals.

- Our nighttime 3-D RTM capability can be applied to simulate nighttime artificial light source emissions as observed from VIIRS and for using in AOD retrievals. Compared to both nighttime and daytime AERONET AODs, a correlation of 0.5-0.6 is found and correlations between VIIRS and AERONET AODs, a result that improves to 0.6-0.7 when observations are confined to higher viewing angles (cosine VZA less than 0.8). This exercise suggests that it is feasible to retrieve nighttime AOD using observations from VIIRS DNB over artificial light sources; however, large uncertainties still exist, possibly due to night-to-night variations in surface light sources emissions and TOA lunar fluxes that need to be carefully quantified in future studies.

- The estimation of surface light emission sources is the key to the nighttime aerosol retrievals using artificial light sources. We found that the NASA Black Marble lunar-BRDF and atmosphere-corrected products (Román et al., 2018) could be used as surface light source emissions for aerosol retrievals, although methods need to be explored for directly using of daily or monthly Black Marble products instead of using yearly averaged monthly data as did in this study.

- Last, to validate the developed 3-D nighttime RTM capability, we inter-compared model simulations between a 1-D RTM (Spherical Harmonic Discrete Ordinate Method for Plane-Parallel Atmospheric Radiative Transfer, or SHDOMPP; Evans, 2007) and 3-D RTM capability developed from this study for both day and nighttime for single column simulations. Those single column simulations are consistent for 1-D and 3-D RTM runs for both day – and night-time simulations, suggesting the nighttime 3-D capability is functioning as designed. Still, for multi-column simulations, much large RMSE error in STD of TOA VIIRS DNB radiance is found for simulations with the 1-D RTM while compared with observations from Dakar, suggesting that adjacency effect needs to be considered for simulation of artificial light source emissions at nighttime.

**Acknowledgements.** We thank the NASA AERONET team for the AERONET data used in this study. We acknowledge the use of imagery from the NASA Worldview application (https://worldview.earthdata.nasa.gov), part of the NASA Earth Observing System Data and Information System (EOSDIS).

**Author contributions.** All authors involved in designing the project. Author J. Z performed the model and data analysis for the project. Author J.S.R provided valuable comments to the study. All authors involved in writing the manuscript.

**Code and data availability:** Both the radiative transfer model mode and data used in the study are publically available. The SHDOM radiative transfer code is freely available from https://coloradolinux.com/shdom/ (last accessed July 14, 2022). The VIIRS-DNB data were downloaded from the NOAA CLASS site (https://www.avl.class.noaa.gov/saa/products/welcome). The NASA's Black Marble data were obtained from NASA LAADS site (https://ladsweb.modaps.eosdis.nasa.gov/). The AERONET data were obtained from the NASA AERONET website (https://aeronet.gsfc.nasa.gov/).

**Competing interests.** The authors claim no competing interests.

**Financial support.** This project is supported by NASA Grant 80NSSC20K1748. Coauthor JSR was supported by the Office of Naval Research Code 322. Author Z. W is also supported by NASA grant 80NSSC22K0199.

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

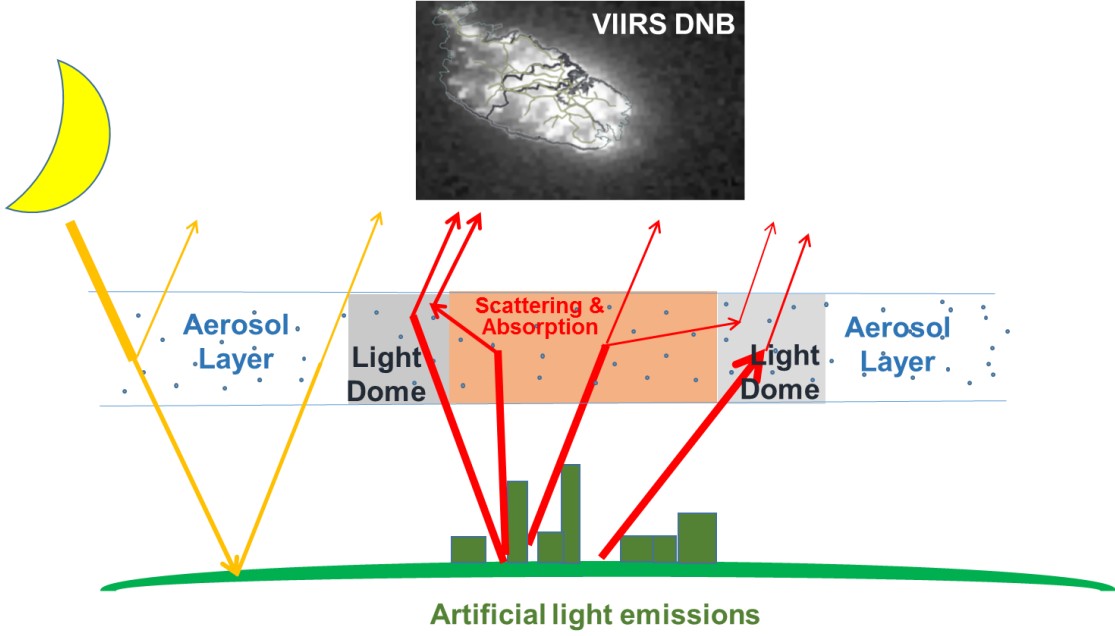

Figure 1. Illustration showing sources of night lights as observed from VIIRS DNB, with Malta included as an example of DNB output (Black Marble Nighttime At Sensor Radiance, July 13, 2021; obtained from NASA Worldview on July 1, 2022; https://worldview.earthdata.nasa.gov/).


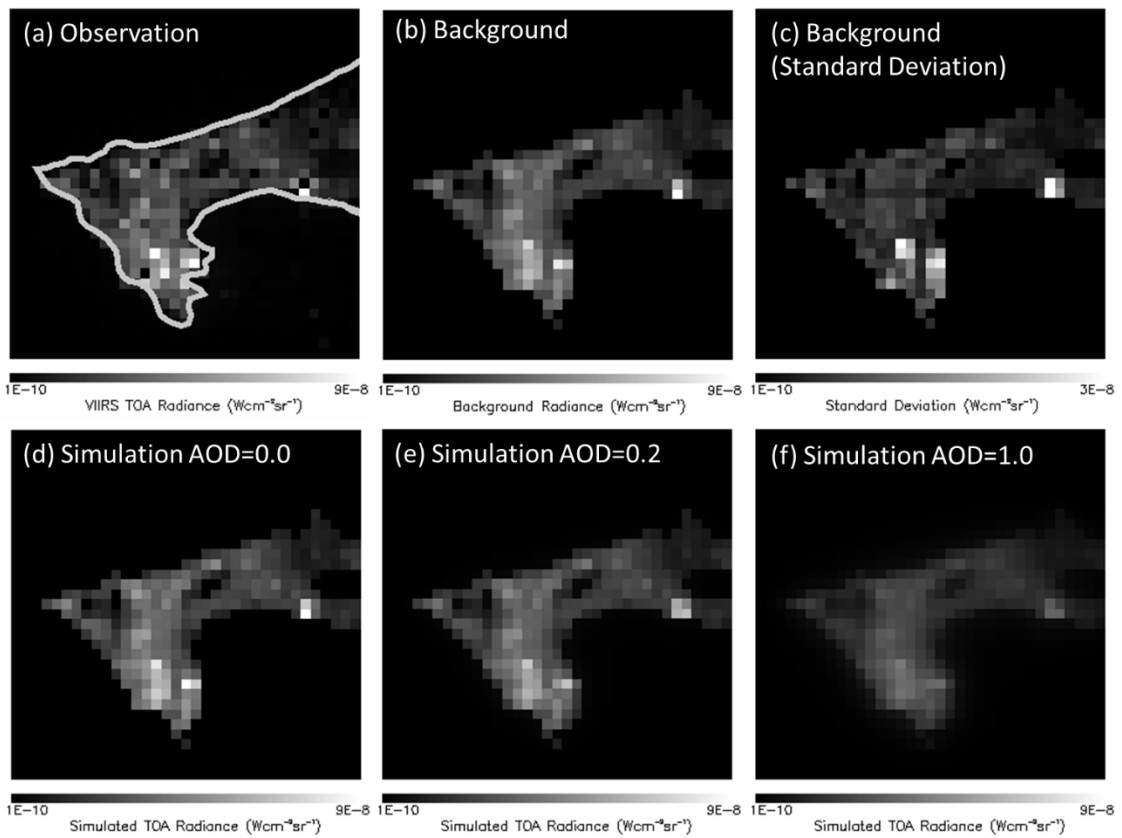


Figure 2a). VIIRS DNB observed TOA radiances on Nov. 28, 2019.  2b). Surface nighttime light emissions from Dakar derived using VIIRS DNB data from 19 nights. 2c) Standard deviation of TOA radiance for data used in generating Fig. 2b.  2d). Simulated TOA radiances for Nov. 18 2019 using 3-D RTM assuming AOD (700 nm) of 0.  2e) Similar to Fig. 1d but for AOD of 0.2.  2e) Similar to Fig. 2d but for AOD of 1.0.



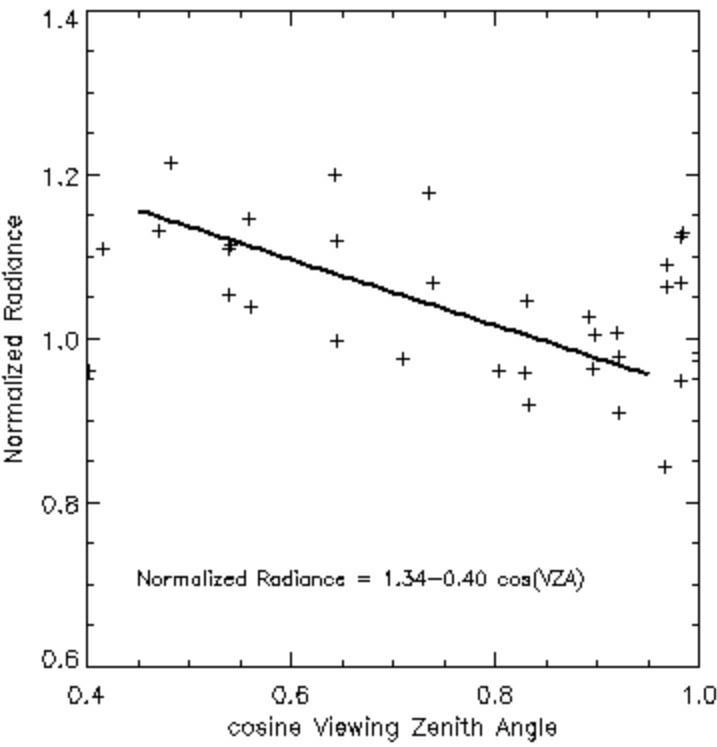

Figure 3. Normalized TOA radiance over Dakar versus cosine VZA. A linear regression line is also shown for data with a
cosine VZA range of 0.45-0.95.


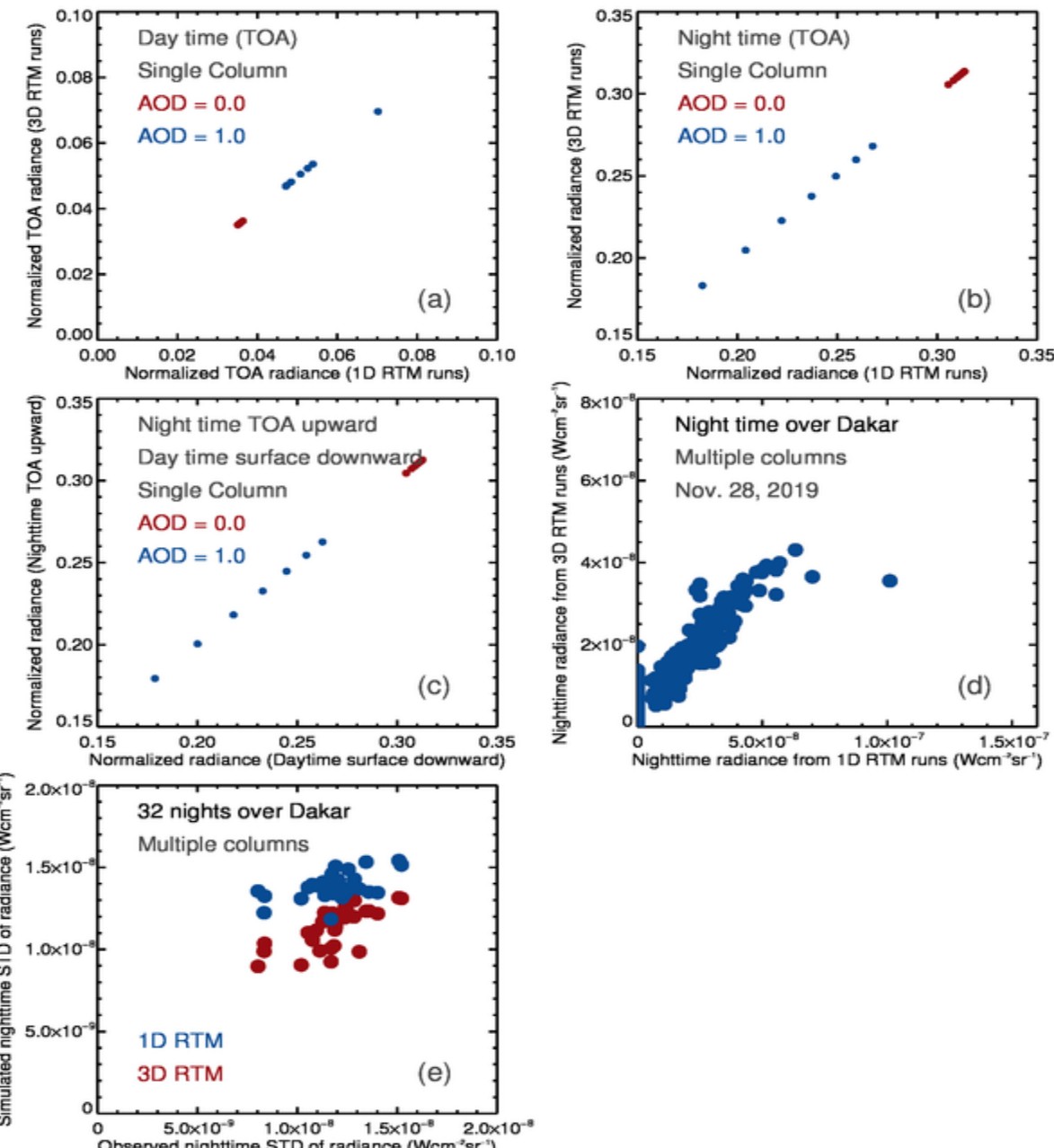

Figure 4a. Simulated day-time normalized TOA radiance from both 1-D and 3-D radiative transfer models for the cosine viewing zenith range of 0.4 to 1.0 and for two aerosol loading scenarios (AOD=0 and 1.0 for AOD at 700 nm), with the TOA downward solar efflux normalized to 1. b). Similar to 4a) but for nighttime. Both sun and lunar fluxes are assumed to be 0 and an artificial light source with normalized Lambertian nighttime artificial emission flux of 1 was used. c). Normalized surface downward flux at surface versus normalized nighttime TOA upward radiance as a function of zenith angle (solar zenith angle for daytime and sensor zenith angle for nighttime) and for 2 aerosol loading as suggested in Figure 4a for a dark surface. At daytime, TOA downward solar flux is normalized to 1. At nighttime surface light emission flux is normalized to 1. d). Comparison of 1-D and 3-D radiative transfer model simulations over Dakar for Nov. 28, 2019 for a study domain of 55x55 grids. e). Comparison of simulated nighttime standard deviations (STD) of radiances and observed nighttime standard deviations (STD) of radiances from VIIRS for 32 nights in 2019 over Dakar.

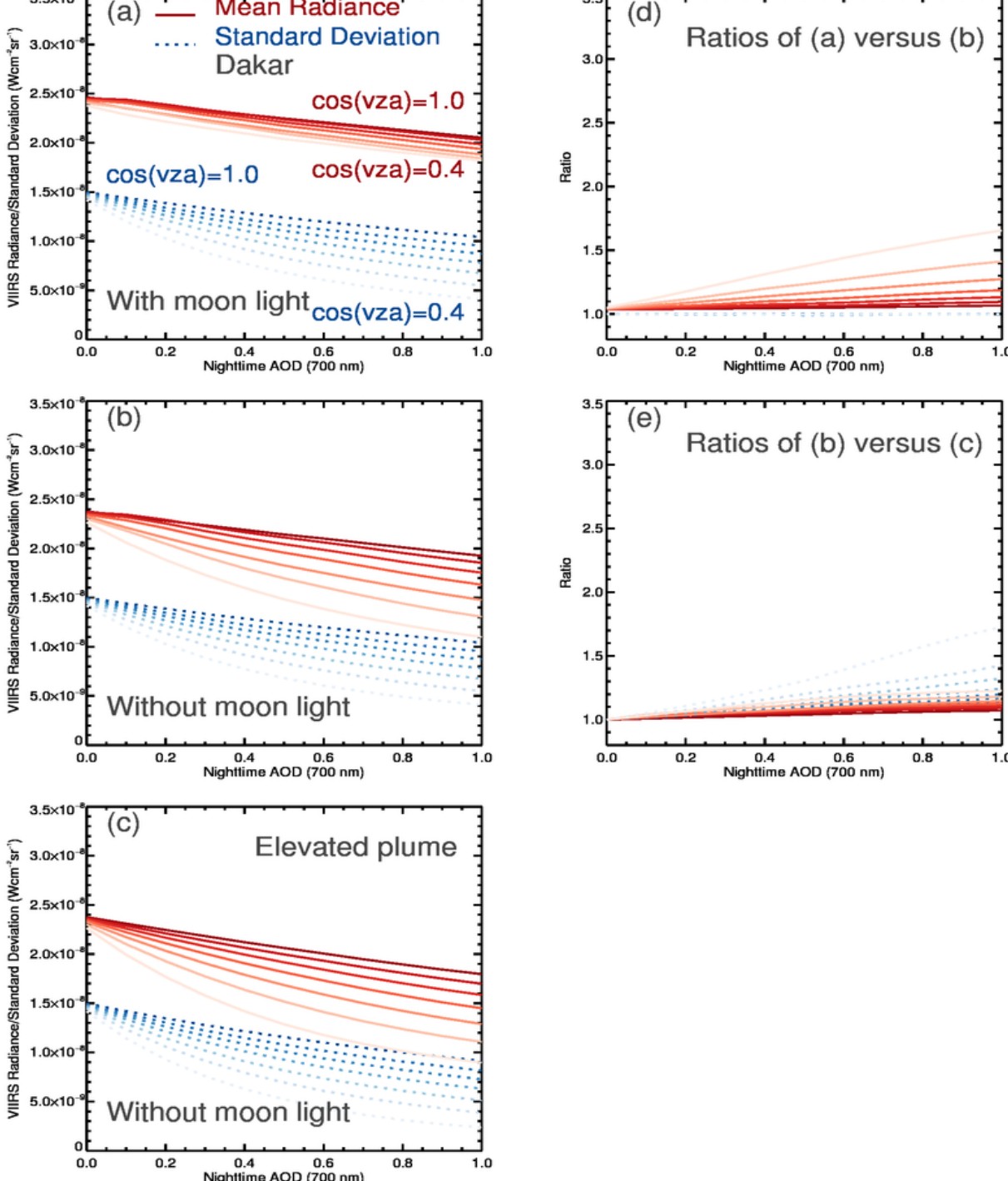

Figure 5a). Simulated mean VIIRS TOA radiance and STD of TOA radiances as a function of nighttime AOD for a moonlit night (TOA downward moon flux of 2.37x10⁻⁸ Wcm⁻² and cosine moon zenith angle of 0.412). 5b). Similar to 5a) but for a moonless night. 5c). Similar to 5b) but for the elevated plume with a peak plume height of 2 km. 5d) Ratios of 5a) versus 5b). 5e) Ratios of 5b) versus 5c)

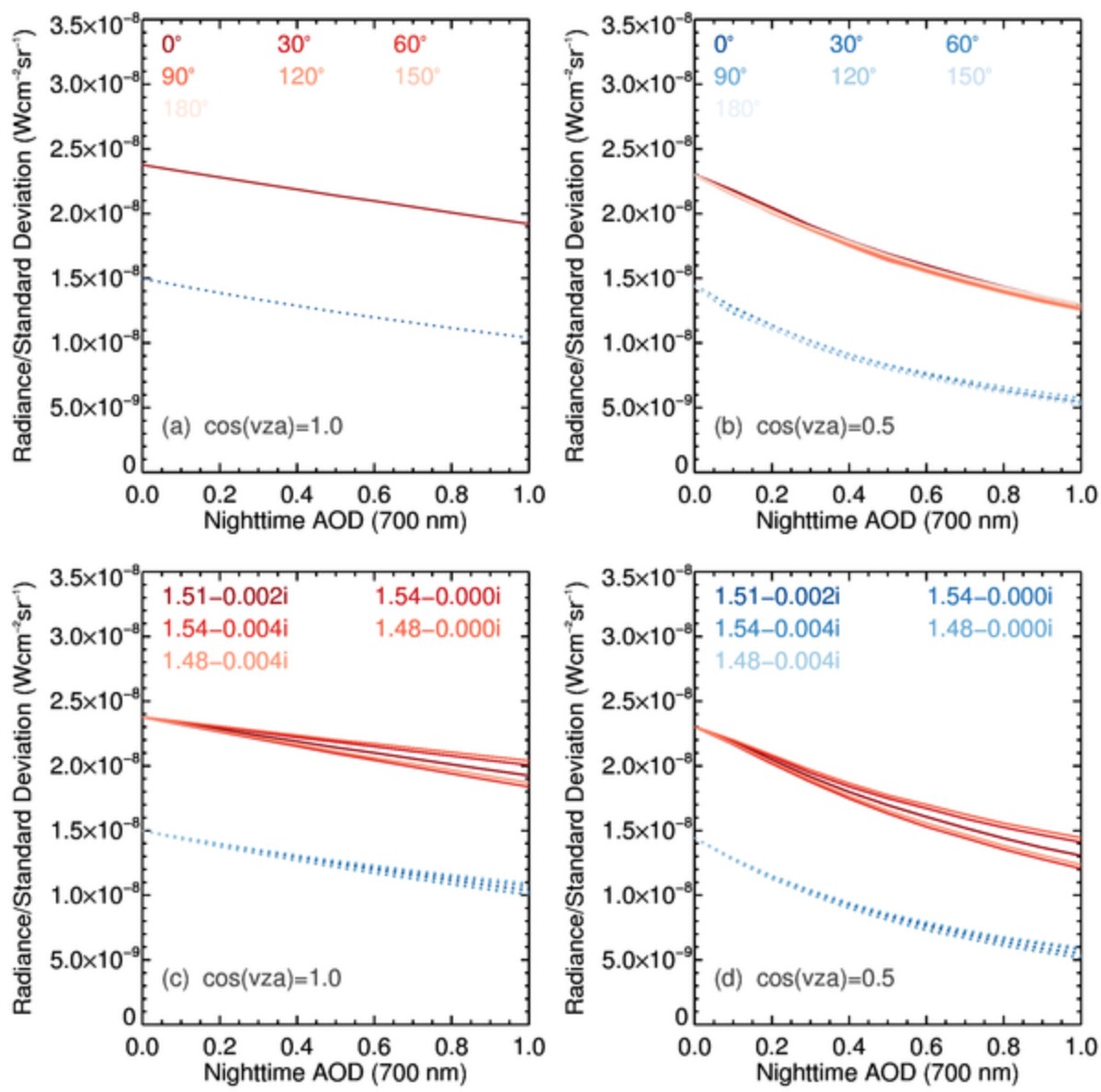


Figure 6a). Simulated mean VIIRS TOA radiance and STD of TOA radiances as a function of nighttime AOD for a moonless night with 7 different azimuth angles of 0°, 30°, 60°, 90°, 120°, 150°, 180° and cosine VZA of 1.0.  6b). Similar to 6a) but for cosine VZA of 0.5.  6c) Simulated mean VIIRS TOA radiance and STD of TOA radiances as a function of nighttime AOD for a moonless night and cosine VZA of 1.0 and with reflective index of dust varying from 1.48-0.000i to 1.54-0.004i.  6d). Similar to
6c) but for cosine VZA of 0.5.

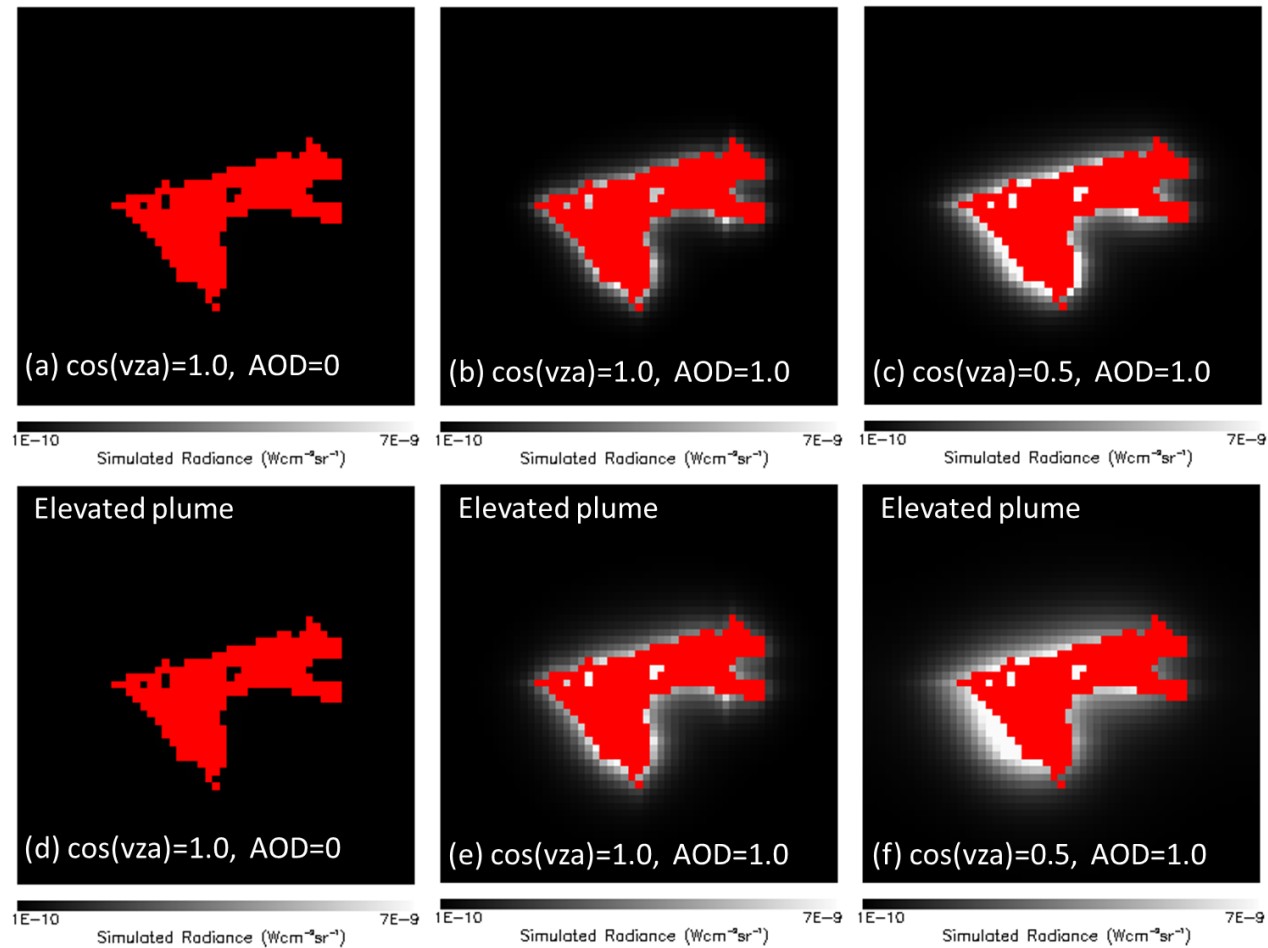

Figure 7a). Simulated TOA radiance over Dakar for a moonless night with cosine VZA of 0 and AOD of 0. Red bins are masked artificial light sources. 7b). Similar to 7a) but for AOD of 1.0. 7c) Similar to 7b) but for cosine VZA of 0.5. 7d-f). Similar to figs. 7a-c but for an elevated plume with the peak altitude of 2 km.


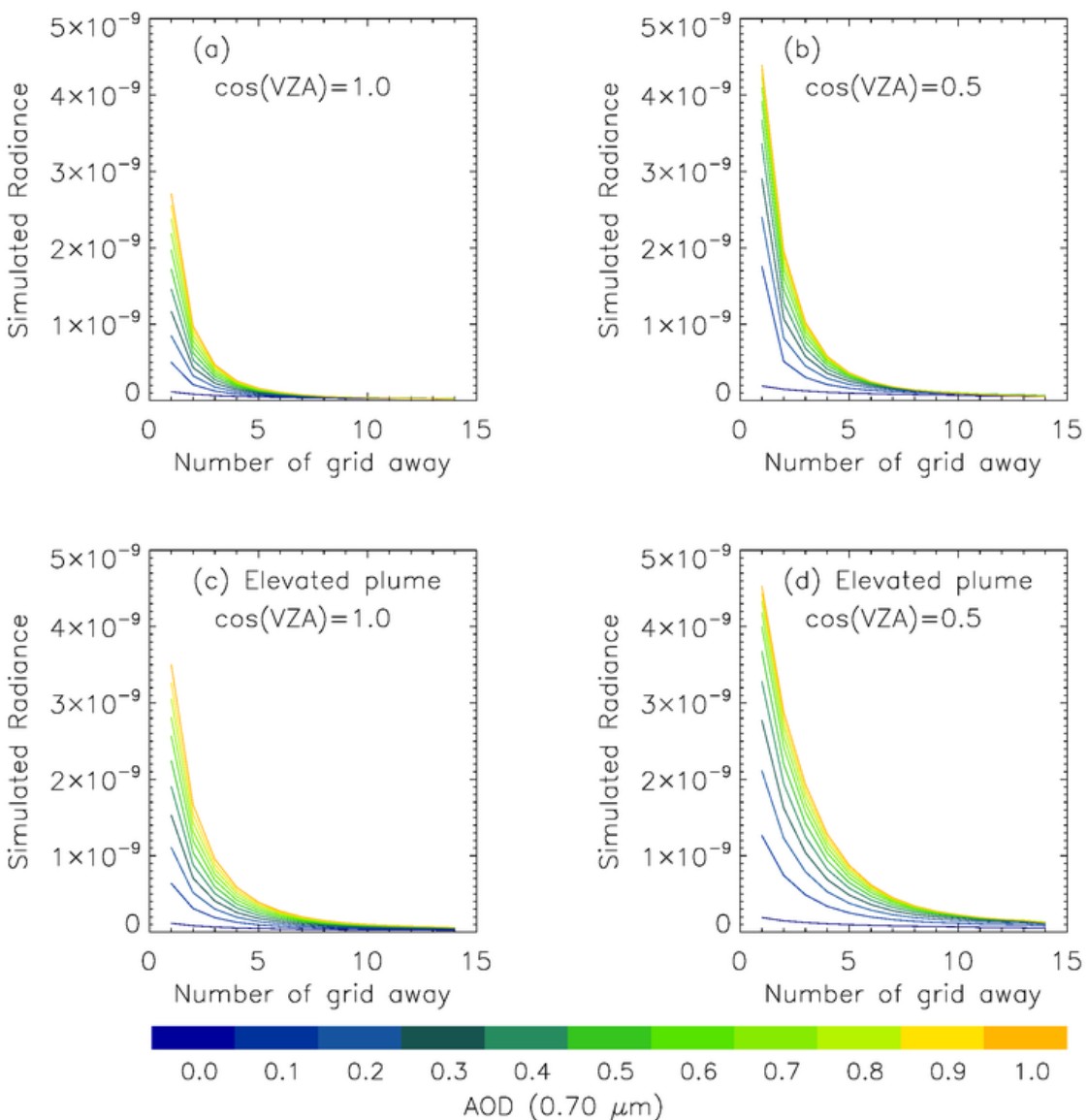

Figure 8a). Simulated VIIRS radiances as a function of distance away from artificial light sources. Cosine VZA is 1.0. 8b) Similar to 8a) but for cosine VZA of 0.5. 8c-d). Similar to figs. 8a-b but for an elevated plume with the peak plume height of 2 km.

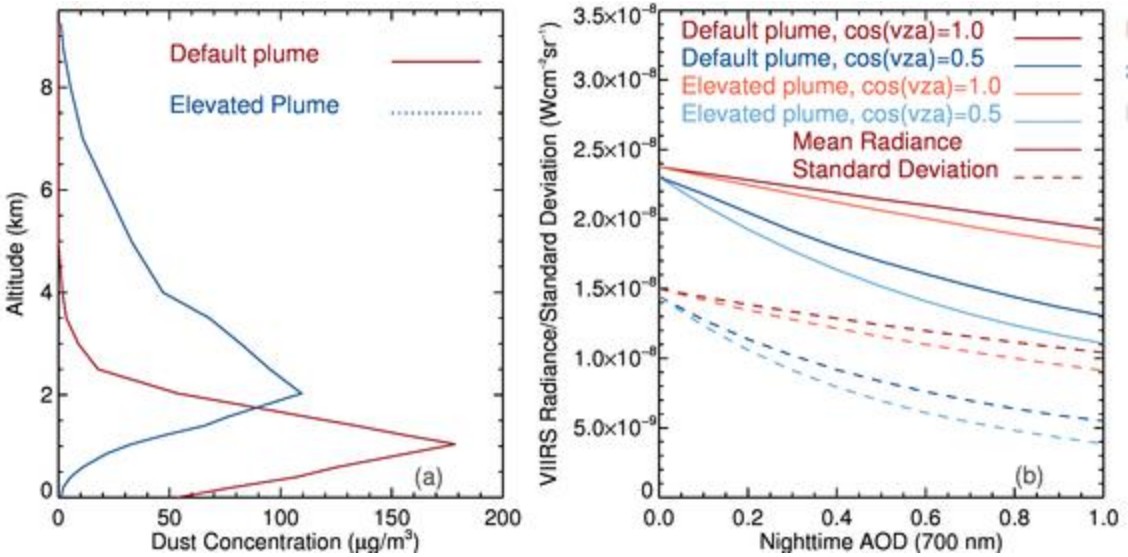

Figure 9a). The red line represents the default vertical distribution of the dust plume used in our study; the peak altitude is at 1 km. The blue line represents an elevated dust plume with a peak altitude of 2 km. In both cases, AOD=0.1. 9b). Simulated mean VIIRS TOA radiance and STD of TOA radiances as a function of nighttime AOD for a moonless night for cosine VZA values of 0.5 and 1.0, for both default and elevated dust plume profiles.


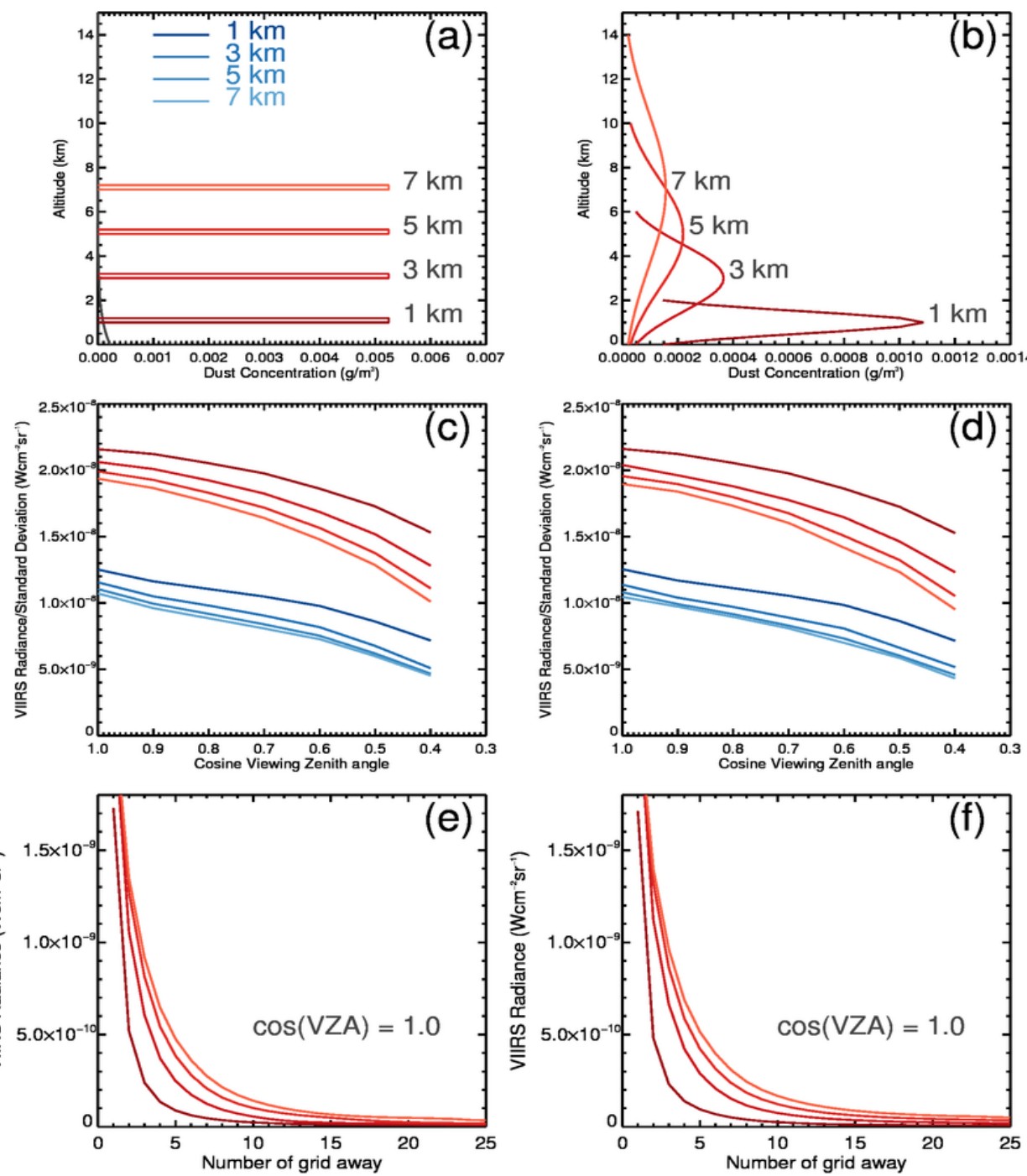


Figure 10a) Aerosol profiles for the first experiment. Two aerosol layers are included, with the first layer (in black) as a background layer with an AOD of 0.1 and the second layer varying from 1 to 3, 5, and 7 km with an AOD of 0.4. 10b) Aerosol
profiles for the second experiment. A single aerosol layer with a Gaussian distribution is included, with peak height varying from 1, 3, 5 and 7 km in altitude. 10c) Simulated VIIRS TOA radiances and STD of radiances for artificial light source regions over Dakar for the first experiment. Red color represents mean radiance and blue color represents the standard deviation of radiance. 10d) Similar to 10c) but for the second experiment. 10e) Simulated VIIRS radiances as a function of distance away from artificial light sources for the nadir view case for the first experiment. 10f). Similar to 10e) but for the second experiment.

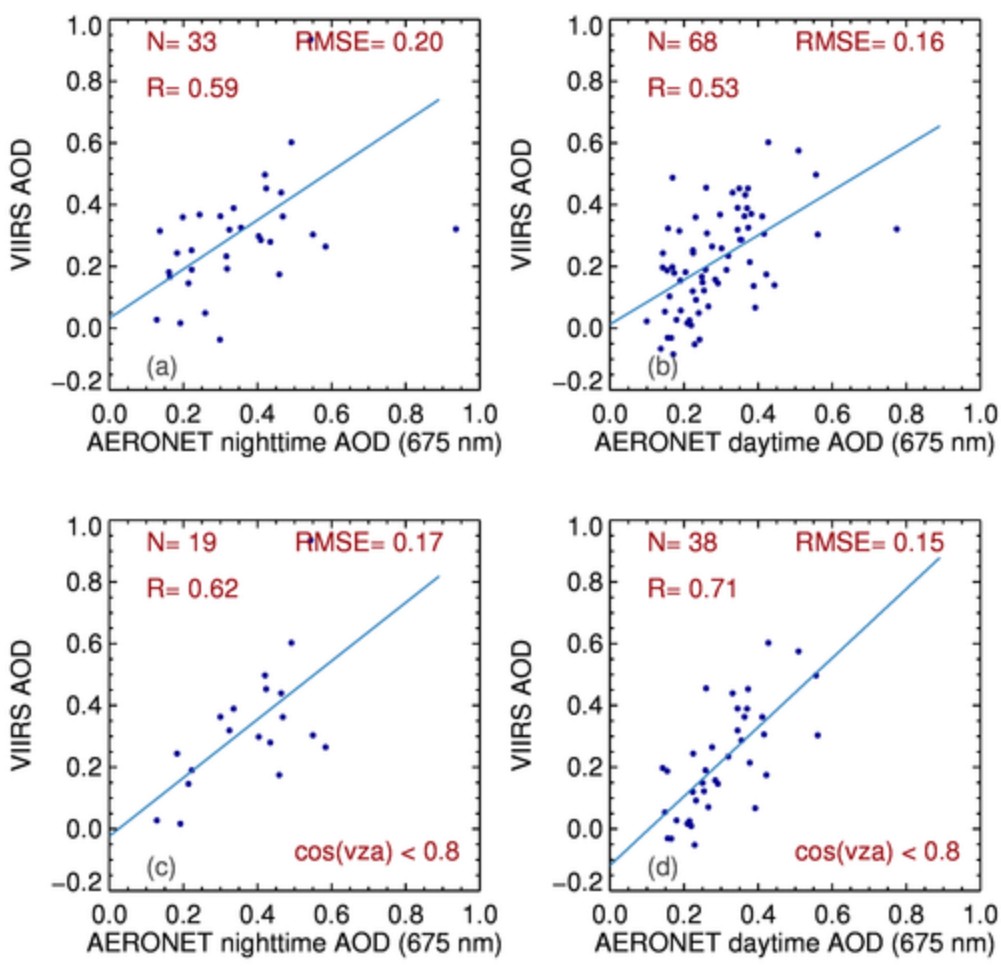


Figure 11a). VIIRS AOD (700 nm) versus nighttime AERONET AOD (675 nm). 11b) VIIRS AOD (700 nm) versus daytime AERONET AOD (675 nm). 11c-d). Similar to figs. 11a-b) but for using VIIRS data with cosine VZA less than 0.8.


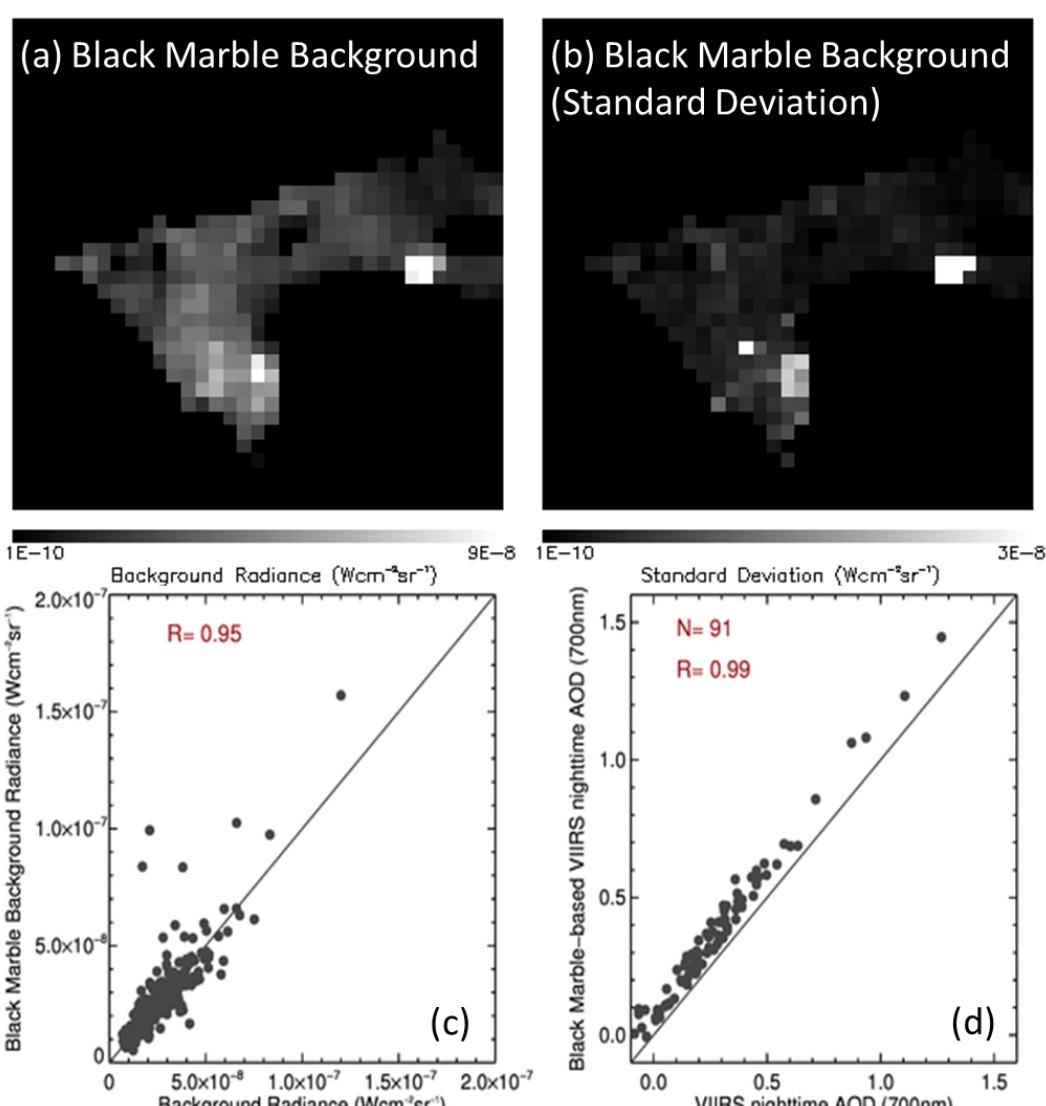

Figure 12a) 12-month averaged clear sky nighttime emissions using the monthly Black Marble lunar BRDF and atmospheric corrected NTL data. 12b). Standard deviations of radiances for data used in generating Fig. 12a. 12c). Scatter plot of surface emissions estimated from this study (Fig. 2a) versus clear sky emissions from Fig. 12a). 12d). Scatter plot of VIIRS AOD retrieved using surface emissions derived from this study versus clear sky artificial light emissions from the Black Marble data as shown in Fig. 12a).