# Peer review of "Sensitivity studies of nighttime TOA radiances from artificial light sources using a 3-D radiative transfer model for nighttime aerosol retrievals"

_Atmospheric Measurement Techniques, 2022_

## Referee Comment (RC1)

General comments

| Principal criteria | Excellent (1) | Good (2) | Fair (3) | Poor (4) |
|---|---|---|---|---|
| **Scientific significance:** Does the manuscript represent a substantial contribution to scientific progress within the scope of Atmospheric Measurement Techniques (substantial new concepts, ideas, methods, or data)? | * | | | |
| **Scientific quality:** Are the scientific approach and applied methods valid? Are the results discussed in an appropriate and balanced way (consideration of related work, including appropriate references)? Note that papers do not necessarily need to be long to be scientifically sound. | | * | | |
| **Presentation quality:** Are the scientific results and conclusions presented in a clear, concise, and well-structured way (number and quality of figures/tables, appropriate use of English language)? | | * | | |

Retrievals of aerosols properties is important for climate, weather, air quality and visibility analyses and forecast. Measurements of atmospheric aerosols during nighttime hours, at large scale from visible satellite data, is a very difficult task because of the low intensity of moon light. The VIIRS Day–Night Band (DNB) onboard the Suomi-NPP satellite is a first-of-its-kind calibrated sensor capable of collecting visible and near-infrared observations during both day and night. In recent studies, both VIIRS DNB observed nighttime light from reflected moon light and from artificial light source emissions were utilized for nighttime aerosol retrieved, by solving 1D radiative transfer (RT) equation. But surface artificial lights have complex properties (3D spatial distribution, non-Lambertian emission,). Therefore, retrieval of aerosol properties under this context can be seen as a true 3D RT problem.

The paper of J. Zhang et al., entitled "Sensitivity studies of nighttime TOA radiances from artificial light sources using a 3-D radiative transfer model for nighttime aerosol retrievals » presents for the first time in my knowledge, a study that investigates the 3D RT problem, in visible wavelength, for atmosphere with (dust) aerosols, with 2D surface artificial light, during the night.

The authors modified the 3D TR model SHDOM (Evans, 1998), included moon light and 2D surface artificial light emission. To my knowledge, this difficult technical work with SHDOM is new. And I don't think this work has ever been done with a Monte Carlo code. But the authors do not explain how they validate their modifications into SHDOM (see may major remarks).

In their work, authors focused over Dakar, used AERONET product (AOD) and the NASA's Black Marble product (surface light source).  By doing 3D TR simulations under different conditions of simulations (zenithal an azimuthal viewing angle, vertical profile of aerosol optical depth), they did sensibility studies for the direct problem (TOA radiances, light dome) and for the inverse problem (aerosol optical depth (AOD)).

They concluded that the STD of TOA radiances is a more stable quantity than mean of TOA radiances for retrieved AOD, that light dome is strong function of aerosol vertical profiles and that light data from NASA's Black Marble product could serve as a primary input into estimation of surface light sources emission. Throughout the text, the authors suggest directions for research (use of light dome, TOA radiances are stronger function of the peak aerosol layer height than of specific shape of aerosol vertical distribution, Black Marble data may need to be revised to better account for viewing zenithal angle dependency,…).

The work of this paper is fundamentally new, and the possibilities of new results is enormous. Globally, I think that results of studies presented in section 3 and 4 are sufficient and represent a good starting point for future research in this topic. Nevertheless, as this new study is based on 3D RT problem, some comparisons with 1D RT problem should be presented (see my major remarks). Otherwise, the text is well written, figures are clear (see my minor remarks).

Major remarks

1) Authors have modified SHDOM and included 2D surface artificial light emission. For example, in annex, it should be nice to see a small study to validate this new code. In 1D it is possible to compare the new code in 1D (IPA – independent pixel approximation, i.e. plane parallel approximation in each column) and a other RT 1D code. By the way, authors could highlight differences between IPA computation and 3D computations with the new code.

2) In section 3 (except for the "light domes" problem), differences between results computed in 1D (IPA approximation that do not take account of "photon horizontal transport" ) and 3D are not presented. We can ask ourselves the question of the importance of taking into account (or not) the 3D aspect of the RT.  It would be good if the authors evaluate a little the differences between calculations of TR in 1D and in 3D.

Minor remarks

1) Line 36 : Cited references do not deal with the climatic aspect.
2) Line 40 : *« For daytime scenarios, where operational aerosol retrievals from reflective solar channels are routinely available from sensors such as Moderate Resolution Imaging Spectroradiometer (MODIS), Multi-angle Imaging SpectroRadiometer (MISR) and Visible Infrared Imaging Radiometer Suite (VIIRS) (e.g. Levy et al., 2013; Hsu et al., 2013; Kahn et al., 2010) ».* This sentence is not clear.
3) Line 65 : a reference is needed.
4) Line 173 : remove semicolon ?
5) Line 190 : what is the model of the Rayleigh scattering. Is there a reference?
6) Line 225 : is it allow to write "Normalized radiance" in a equation ?
7) Line 305 and Fig 7 and 9 : Wouldn't it be better to express the abscissa in length (m) rather than in number of pixels ?
8) Line 318 : « As it is difficult for 2-D RTMs to accurately account for scattered lights originated outside the targeted 2-D domain ». This sentence is incomplete ?
9) Line 378 : I don't understand why the paper of Jakel et al. (2013) is relevant of the value of 0.1 ?
10) Line 583 : « showing sources »
11) Figure 8a : Wouldn't it be better to express the abscissa in extinction (m-1) rather concentration ?
12) Figure 9 : Please give explanation about the red and blue color.

---

## Author Comment (AC1)

*Retrievals of aerosols properties is important for climate, weather, air quality and visibility analyses and forecast. Measurements of atmospheric aerosols during nighttime hours, at large scale from visible satellite data, is a very difficult task because of the low intensity of moon light. The VIIRS Day–Night Band (DNB) onboard the Suomi-NPP satellite is a first-of-its-kind calibrated sensor capable of collecting visible and near-infrared observations during both day and night. In recent studies, both VIIRS DNB observed nighttime light from reflected moon light and from artificial light source emissions were utilized for nighttime aerosol retrieved, by solving 1D radiative transfer (RT) equation. But surface artificial lights have complex properties (3D spatial distribution, non-Lambertian emission,). Therefore, retrieval of aerosol properties under this context can be seen as a true 3D RT problem.*

*The paper of J. Zhang et al., entitled "Sensitivity studies of nighttime TOA radiances from artificial light sources using a 3-D radiative transfer model for nighttime aerosol retrievals » presents for the first time in my knowledge, a study that investigates the 3D RT problem, in visible wavelength, for atmosphere with (dust) aerosols, with 2D surface artificial light, during the night.*

*The authors modified the 3D TR model SHDOM (Evans, 1998), included moon light and 2D surface artificial light emission. To my knowledge, this difficult technical work with SHDOM is new. And I don't think this work has ever been done with a Monte Carlo code. But the authors do not explain how they validate their modifications into SHDOM (see may major remarks).*
*In their work, authors focused over Dakar, used AERONET product (AOD) and the NASA's Black Marble product (surface light source). By doing 3D TR simulations under different conditions of simulations (zenithal an azimuthal viewing angle, vertical profile of aerosol optical depth), they did sensibility studies for the direct problem (TOA radiances, light dome) and for the inverse problem (aerosol optical depth (AOD)). They concluded that the STD of TOA radiances is a more stable quantity than mean of TOA radiances for retrieved AOD, that light dome is strong function of aerosol vertical profiles and that light data from NASA's Black Marble product could serve as a primary input into estimation of surface light sources emission. Throughout the text, the authors suggest directions for research (use of light dome, TOA radiances are stronger function of the peak aerosol layer height than of specific shape of aerosol vertical distribution, Black Marble data may need to be revised to better account for viewing zenithal angle dependency,...).*

*The work of this paper is fundamentally new, and the possibilities of new results is enormous. Globally, I think that results of studies presented in section 3 and 4 are sufficient and represent a good starting point for future research in this topic. Nevertheless, as this new study is based on 3D RT problem, some comparisons with 1D RT problem should be presented (see my major remarks). Otherwise, the text is well written, figures are clear (see my minor remarks).*

We thank the reviewer for the constructive comments.  The reviewer is correct that we need to validate the 3-D model with a 1-D model.  In fact, we also realized the need for a 1-D and 3-D model comparison and conducted the study this past winter.  Through this process, we found and

corrected a bug that impacts some figures and has a marginal impact on discussions (we updated discussions and figures in the revised version of the paper). We contacted AMT and were instructed to include changes in the revised version of the paper. We thus added 1-D and 3-D RTM comparison to this revised version of the paper.

*Major remarks*
*1) Authors have modified SHDOM and included 2D surface artificial light emission. For example, in annex, it should be nice to see a small study to validate this new code. In 1D it is possible to compare the new code in 1D (IPA – independent pixel approximation, i.e. plane parallel approximation in each column) and a other RT 1D code. By the way, authors could highlight differences between IPA computation and 3D computations with the new code.*

This is an excellent suggestion. We also realized the need for a 1-D and 3-D comparison and conducted the study last winter (per suggestion from AMT, we included the changes in this version of the revised paper). The 1-D Spherical Harmonic Discrete Ordinate Method for Plane-Parallel Atmospheric Radiative Transfer (SHDOMPP, Evans, 2007) was chosen for this purpose. Note that SHDOMPP was previously validated against the Discrete Ordinate Radiative Transfer model (DISORT) in a past study for daytime applications (Evans, 2007). We chose the SHDOMPP 1-D model as it takes similar input parameters as the 3-D RTM used in this study, and parameter conversion-related issues could be minimized. By running the 3D-RTM over a single grid, we inter-compared the 3-D model against the 1-D SHDOMPP model for the following day and nighttime cases.

1. We inter-compared simulated single column TOA radiance at daytime from both the 1-D and 3-D RTMs.
2. We enhanced the 1-D SHDOMPP model with nighttime capabilities by adding surface light emission as a lower boundary condition and replaced TOA downward solar radiation with TOA downward moon light. We then inter-compared the single column simulations at nighttime from both 1-D and 3-D RTMs.
3. To further validate the developed nighttime 3-D RTM capability, for a given zenith angle (solar zenith angle for the TOA downward path and sensor zenith angle for the surface upward path), we inter-compared daytime surface downward radiances (assuming normalized TOA input solar flux of 1, with no surface artificial light sources and a dark surface) and nighttime TOA upward radiances (assuming normalized surface light source emission flux of 1, with no TOA solar or moon flux and a dark surface) from the single column 3-D RTM runs. This exercise ensures that for a given zenith angle and emission source, outgoing radiance from the upward path from the nighttime RTM and surface reaching radiance from the downward path from the daytime RTM match as they both go through the similar radiative process. Also given that the daytime process (daytime downward path) has already been validated against the 1-D RTM, this process ensures that nighttime simulation are also functioning as designed.

Further, we incorporated the validated efforts in the text (Sect. 2.6) as suggested by the reviewer.

*2) In section 3 (except for the "light domes" problem), differences between results computed in 1D (IPA approximation that do not take account of "photon horizontal transport") and 3D are not presented. We can ask ourselves the question of the importance of taking into account (or not) the 3D aspect of the RT. It would be good if the authors evaluate a little the differences between calculations of TR in 1D and in 3D.*

This is another excellent suggestion. We actually also conducted the study last winter during the review period as well. In the exercise, both the 1-D (with nighttime capability enhanced as mentioned above) and 3-D radiative transfer models were also applied to simulate VIIRS nighttime radiances over Dakar for multi-column simulations for a study domain with 3025 (55x55) grid points. We included the study in the text in Section 2.6.

*Minor remarks*
*1) Line 36 : Cited references do not deal with the climatic aspect.*

We included two references as suggested.

Kaufman, Y., Tanré, D. and Boucher, O.: A satellite view of aerosols in the climate system, Nature, 419, 215–223, https://doi.org/10.1038/nature01091, 2002.

Alfaro-Contreras R., Zhang J., Reid J. S., and Christopher S.: A study of 15-year aerosol optical thickness and direct shortwave aerosol radiative effect trends using MODIS, MISR, CALIOP and CERES, Atmos. Chem. Phys., 17, 13849-13868, https://doi.org/10.5194/acp-17-13849-2017, 2017.

*2) Line 40 : « For daytime scenarios, where operational aerosol retrievals from reflective solar channels are routinely available from sensors such as Moderate Resolution Imaging Spectroradiometer (MODIS), Multi-angle Imaging SpectroRadiometer (MISR) and Visible Infrared Imaging Radiometer Suite (VIIRS) (e.g. Levy et al., 2013; Hsu et al., 2013; Kahn et al., 2010) ». This sentence is not clear.*

Thanks for pointing out the issue. We rewrote the sentence to "For daytime scenarios, operational aerosol retrievals from reflective solar channels are routinely available from sensors such as Moderate Resolution Imaging Spectroradiometer (MODIS), Multi-angle Imaging SpectroRadiometer (MISR) and Visible Infrared Imaging Radiometer Suite (VIIRS) (e.g. Levy et al., 2013; Hsu et al., 2013; Kahn et al., 2010)."

*3) Line 65 : a reference is needed.*

We added references.

Johnson R. S., Zhang J., Reid J. S., Hyer E. J., and Miller S. D.: Toward Nighttime Aerosol Optical Depth Retrievals from the VIIRS Day/Night Band, Atmos. Meas. Tech., 6, 1245-1255, doi:10.5194/amt-6-1245-2013, 2013.

McHardy T., Zhang J., Reid J. S., Miller S. D., Hyer E. J., and Kuehn R.: An improved method for retrieving nighttime aerosol optical thickness from the VIIRS Day/Night Band, Atmos. Meas. Tech., 8, 4773-4783, doi:10.5194/amt-8-4773-2015, 2015.

Zhang, J., Jaker, S. L., Reid, J. S., Miller, S. D., Solbrig, J., and Toth, T. D.: Characterization and application of artificial light sources for nighttime aerosol optical depth retrievals using the Visible Infrared Imager Radiometer Suite Day/Night Band, Atmos. Meas. Tech., 12, 3209–3222, https://doi.org/10.5194/amt-12-3209-2019, 2019.

*4) Line 173 : remove semicolon ?*

Done.

*5) Line 190 : what is the model of the Rayleigh scattering. Is there a reference?*

Yes, we added references.

Evans, K. F.: The spherical harmonics discrete ordinate method for three-dimensional atmospheric radiative transfer. *J. Atmos. Sci.*, **55,** 429–446, 1998.

Fu, Q., and K. N. Liou K. N.: On the correlated *k*-distribution method for radiative transfer in nonhomogeneous atmospheres, J. Atmos. Sci., 49, 2139–2156, 1992.

6) Line 225 : is it allow to write "Normalized radiance" in a equation ?

We are unsure. We defined normalized radiance as "$I_{normalized}$" and changed the equation to

$I_{normalized} = 1.34 - 0.40 \cos (VZA)$.

*7) Line 305 and Fig 7 and 9 : Wouldn't it be better to express the abscissa in length (m) rather than in number of pixels ?*

Since the grids are equal in length, number of pixels/grid points and length essentially refer to the same thing. Thus, we think it is a personal preference. We still like to use number of pixels/grid points as it is a directly traceable parameter from the model.

*8) Line 318 : « As it is difficult for 2-D RTMs to accurately account for scattered lights originated outside the targeted 2-D domain ». This sentence is incomplete ?*

We rewrote the sentence as "This is because it is difficult for 2-D RTMs to accurately account for scattered lights originated outside the targeted 2-D domain".

*9) Line 378 : I don't understand why the paper of Jakel et al. (2013) is relevant of the value of 0.1 ?*

In Figure 4a of the paper, observed, area-averaged surface albedo for land is shown as a function of wavelength, and the value is close to 0.1 near 700nm.

*10) Line 583 : « showing sources »*

Corrected.  Thanks.

*11) Figure 8a : Wouldn't it be better to express the abscissa in extinction (m-1) rather concentration ?*

Extinction can be converted from concentration by multiplying by the mass extinction coefficient.  Thus, we think both parameters are reasonable to be used and it is a rather personal preference.  We chose concentration as it is the input parameter for the model, and rather straightforward to use.

*12) Figure 9 : Please give explanation about the red and blue color*

Thanks for pointing out the issue.  Red color represents mean radiance and blue color represents the standard deviation of radiance.  We added the explanation in the figure caption.

---

## Author Comment (AC2)

*The article by Zhang et al. investigates the impact of various parameters on nighttime satellite aerosol retrievals using surface artificial light emission sources and a 3-D radiative transfer model, following their earlier work, Zhang et al., 2008. The draft reads technically sound with good analysis and its context is logically organized. The introduction also provides enough background for readers and cites necessary previous works. Therefore, I recommend publishing after addressing some minor comments:*

We thank the reviewer for their time and the constructive comments provided.

*L54, can you provide a few more words to explain what is "retrieval-filled value issue"?*

We added this sentence: "for which aerosol signals as received from lidars are too low for retrievals and thus retrieval filled values are assigned that may introduce sampling-related biases "

*L93, since the spatial resolution is highly variable as mentioned here, does this paper discuss the impact of the spatial resolution on nighttime retrievals in the later context?*

We didn't include the mentioned discussions in this study as we used VIIRS data that have a known spatial resolution.  Still, this is an interesting research topic and can be a full research topic of its own.  This is especially true because, to carefully resolve the issue, observations from different platforms with different spatial resolutions are needed for validation/evaluation efforts. Thus, we leave this question for a future study.

*L160, what is the distance threshold for choosing the pairs?*

Since AERONET data from the Dakar site (14.394°N, 16.959°W) were used for evaluating retrievals from Dakar, there is no need for a distance threshold.  The AERONET site is located within the city.

*Figure 1 caption, add space between "showing" and "sources". I feel this figure can be improved. I don't see the "sensor" or satellite on this figure. Is it helpful to label come concepts such as VZA in the figure?*

We added a space between "showing" and "resources".  Also, we added "VIIRS DNB" in Figure 1 as suggested.  We prefer not to label VZA as we need to add an assisting line to introduce the concept of VZA, which will make the overall structure of Figure 1 less ideal.  Also, VZA is a fundamental parameter in remote sensing, and thus we decided to not draw VZA in Figure 1.

*L372, how many AERONET sites for their data are involved in this study?*

Only one; the Dakar AERONET site was used as we conducted the study over Dakar

*I feel the following article is relevant and should be cited. Cavazzani, S., Ortolani, S., Bertolo, A., Binotto, R., Fiorentin, P., Carraro, G., & Zitelli, V. (2020). Satellite measurements of artificial light at night: Aerosol effects. Monthly Notices of the Royal Astronomical Society, 499(4), 5075-5089.*

This reference has been added.